# Orodispersible Films: Current Innovations and Emerging Trends

**DOI:** 10.3390/pharmaceutics15122753

**Published:** 2023-12-11

**Authors:** Shery Jacob, Sai H. S. Boddu, Richie Bhandare, Samiullah Shabbir Ahmad, Anroop B. Nair

**Affiliations:** 1Department of Pharmaceutical Sciences, College of Pharmacy, Gulf Medical University, Ajman P.O. Box 4184, United Arab Emirates; samiullah@gmu.ac.ae; 2Department of Pharmaceutical Sciences, College of Pharmacy and Health Sciences, Ajman University, Ajman P.O. Box 346, United Arab Emirates; s.boddu@ajman.ac.ae (S.H.S.B.); r.bhandareh@ajman.ac.ae (R.B.); 3Center of Medical and Bio-Allied Health Sciences Research, Ajman University, Ajman P.O. Box 346, United Arab Emirates; 4Department of Pharmaceutical Sciences, College of Clinical Pharmacy, King Faisal University, Al-Ahsa 31982, Saudi Arabia; anair@kfu.edu.sa

**Keywords:** orodispersible film, polymers, manufacturing methods, 3D printing, clinical trials, patents, evaluation

## Abstract

Orodispersible films (ODFs) are thin, mechanically strong, and flexible polymeric films that are designed to dissolve or disintegrate rapidly in the oral cavity for local and/or systemic drug delivery. This review examines various aspects of ODFs and their potential as a drug delivery system. Recent advancements, including the detailed exploration of formulation components, such as polymers and plasticizers, are briefed. The review highlights the versatility of preparation methods, particularly the solvent-casting production process, and novel 3D printing techniques that bring inherent flexibility. Three-dimensional printing technology not only diversifies active compounds but also enables a multilayer approach, effectively segregating incompatible drugs. The integration of nanoparticles into ODF formulations marks a significant breakthrough, thus enhancing the efficiency of oral drug delivery and broadening the scope of the drugs amenable to this route. This review also sheds light on the diverse in vitro evaluation methods utilized to characterize ODFs, ongoing clinical trials, approved marketed products, and recent patents, providing a comprehensive outlook of the evolving landscape of orodispersible drug delivery. Current patient-centric approaches involve developing ODFs with patient-friendly attributes, such as improved taste masking, ease of administration, and enhanced patient compliance, along with the personalization of ODF formulations to meet individual patient needs. Investigating novel functional excipients with the potential to enhance the permeation of high-molecular-weight polar drugs, fragile proteins, and oligonucleotides is crucial for rapid progress in the advancing domain of orodispersible drug delivery.

## 1. Introduction

Oral medications are the preferred and widely accepted method of drug delivery due to their ease of administration, convenience for repeated and prolonged use, non-invasiveness, adaptability, scalability, and high patient compliance [1]. Nevertheless, certain patient groups, such as the elderly, children, individuals with Parkinson’s disease, and those recovering from anesthesia, often encounter challenges in swallowing or chewing solid dosage forms, particularly tablets and capsules [2]. In the United States, it is estimated that about 15 million people have dysphagia, and this represents about 4.6% of the population. Thus, extensive efforts have been undertaken to create innovative oral drug formulations that dissolve or disperse in the oral cavity, aiming to address the issue of swallowing difficulties [3]. Moreover, highly vascularized oral mucosa may increase permeability to many medications, thereby providing a rapid onset of action and increasing bioavailability, as reported elsewhere [4].

Oral disintegrating tablets (ODTs) are a type of solid oral dosage form designed to disintegrate rapidly within a matter of seconds when placed on the tongue without the necessity of water or chewing [5]. Oral thin films are a drug delivery system consisting of thin, flexible sheets that typically dissolve or disintegrate quickly, often within seconds, when placed in the mouth. They are intended to be placed either on the tongue or cheek and can be used to deliver a variety of medications, including over-the-counter (OTC) and prescription drugs [6,7].

While various names like thin strip, oral film, orally dissolving film, quick dissolve film, melt-away film, and wafer are employed to refer to the oral film dosage form, the European Medicines Agency officially designates it as an orodispersible film (ODF), or, as the United States Food and Drug Administration (U.S. FDA) commonly terms them, soluble films [8]. As per European Pharmacopeia (Ph. Eur.), ODFs are defined as sheets, either single or multilayered, composed of appropriate materials and are intended for rapid dispersion in the mouth. They rapidly disintegrate/dissolve in saliva to form a solution or suspension, thus enabling rapid absorption and delivery of the drug into the bloodstream or a rapid local effect. Moreover, ODFs offer rapid and consistent drug release, which can improve the bioavailability of some medications. The oral cavity is richly vascularized and has low enzymatic activity, which can potentially boost the bioavailability of drugs with low aqueous solubility. This route is advantageous for those drugs classified under the biopharmaceutical classification system (BCS) as Class II and Class IV. Rapid permeation across the mucosal lining of the oral cavity can circumvent acid hydrolysis in the stomach and initial hepatic metabolism. This pathway is particularly well-suited for potent medications, especially those designed for acute conditions, where they have an immediate therapeutic effect, mainly due to oromucosal and pregastric absorption, as well as direct access to the jugular vein [4]. Nevertheless, certain compounds are absorbed exclusively in the gastrointestinal tract after ingestion.

This review aims to comprehensively explore the current innovations and emerging trends of ODFs. It focuses on their design, formulation components, and manufacturing methods for potential applications in oral drug delivery. The objective is to provide a thorough understanding of ODF versatility, the recent advancements in polymers and plasticizers, and the integration of novel techniques like 3D printing. Additionally, the review seeks to emphasize the impact of incorporating nanoparticles in ODF formulations for enhanced oral drug delivery efficiency. This review adopts a systematic approach, conducting comprehensive literature searches on databases such as PubMed, Scopus, Web of Science, clinical trial databases, and patent databases. It focuses on keywords like ‘orodispersible film’, ‘polymers’, ‘manufacturing methods’, ‘3D printing’, ‘clinical trials’, ‘patents’, and ‘evaluation’. The inclusion and exclusion criteria ensure the selection of reliable studies and patents. Structured data extraction, critical appraisal, and synthesis of information are employed to present a cohesive narrative on the present advancements and evolving trends in orodispersible films while considering database-specific functionalities and intricacies.

The ODF, or strip, typically employs a hydrophilic polymer, preferably with mucoadhesive properties, and aims to achieve rapid disintegration within a short period. The primary driving factors responsible for the rapid drug release of ODFs, as anticipated, are the hydration and subsequent swelling of polymers due to water diffusion [9]. The primary objective of the majority of ODFs is to quickly dissolve or disintegrate in the oral cavity, forming a solution or suspension, which is then swallowed for absorption in the gastrointestinal tract [10].

The primary constraints frequently encountered with oral films pertain to their vulnerability in high-humidity environments and their limited capacity to accommodate a substantial drug dosage. Table 1 illustrates a comparison between the advantages and drawbacks of orodispersible film drug delivery systems. The limited formulation size restricts the inclusion of additives for taste masking the drug, potentially impacting patient compliance.

## 2. Formulation Components

The components of oral films can differ depending on the particular formulation and intended purpose. However, there are common constituents typically found in oral film formulations, such as active pharmaceutical ingredients (APIs), polymers, plasticizers, sweetening agents, flavoring agents, coloring agents, salivary stimulating agents, stabilizers, surfactants, and solvents. In the context of pharmaceutical ODF dosage form development, critical quality attributes (CQAs) comprise the physical, chemical, biological, or microbiological properties and characteristics that must fall within specified limits to ensure the desired quality of the product. These attributes can be influenced by essential material qualities, including the quality of the API and the physicochemical properties of excipients, as well as critical process parameters such as the order of adding raw materials, the sequence of adding solutions, and the temperature of water [11]. Table 2 illustrates a generic critical quality attribute for an orodispersible film.

### 2.1. APIs

APIs in ODFs can vary depending on the intended therapeutic purpose, and the selection depends on the drug’s solubility, stability, and desired pharmacokinetic profile. Currently, ODFs have been investigated with drugs belonging to different therapeutic categories, including anti-allergics and antihistamines, antibiotics, anti-inflammatories and analgesics, antidiarrheals, antiemetics, antidepressants, antipsychotics, anticonvulsants, cardiovascular medications, neurological medications, vitamins, and nutritional supplementation [12]. APIs are typically incorporated (5% *w*/*w* to 30% *w*/*w*) either as a solid dispersion, nanosuspension, micronized powder, or nanocrystals within the film matrix.

ODFs predominantly consist of BCS class I drugs, chosen primarily for their high solubility and permeability characteristics, although drugs from other classes are also integrated. The weight of the ODF, being less than 200 mg, limits the formulation space for the API, restricting its application to potent active ingredients. Notably, a higher molecular weight film former tends to yield a greater sample, dependent on its concentration. Due to the limited space within the oral cavity, ODFs with a size of 2 × 2 cm^2^ and a maximum thickness of 100 µm are considered acceptable [13]. During the polymer mixing process, several issues, such as the entrapment of air bubbles and the use of inappropriate casting solution viscosities, can affect the drug’s stability and dose uniformity. Moreover, the utilization of organic solvents during the process and the existence of residual solvents in the dried sample can potentially lead to toxicity issues and raise regulatory concerns [8]. 

### 2.2. Polymers

Oral films primarily comprise a polymer matrix, which provides film-forming properties with structural integrity. A range of polymers, including but not limited to starch, modified starches, hydroxypropyl methylcellulose (HPMC) (including hypromellose variants E3, E5, and E15), sodium carboxymethyl cellulose (NaCMC), gelatin, hydroxypropyl cellulose (HPC), hydroxyethyl cellulose (HEC), pectin, carboxymethyl cellulose (CMC), pullulan, locust bean gum, xanthan gum, guar gum, carrageenan, povidone polymers (polyvinylpyrrolidone, PVP), polyvinyl alcohol (PVA), polyethylene oxide (PEO), maltodextrins (MDXs), and various others have been studied as potential base materials for producing ODFs [8]. 

The film-forming polymer, which serves as the major component of the ODF, constitutes up to 65% of the weight based on the total dry weight of the film [14]. In some cases, a combination of polymers is employed to enhance the hydrophilicity, flexibility, mouthfeel attributes, and solubility of ODFs. These polymers should be non-toxic, non-irritating, free from leachable impurities, possess excellent wetting and spreadability, have a good shelf life, and should not promote secondary infections in the oral mucosa or dental regions. Additionally, the ODFs should have adequate peel, shear, and tensile strength. Among water-soluble polymers, gelatin, and hypromellose are the most commonly used to prepare oral strips. Table 3 shows a comprehensive overview of the frequently used film-forming polymers, plasticizers, APIs, preparation methods, and the key highlights related to ODFs. The acceptability of ODFs depends upon film composition and the formation process, which affects disintegration, taste, texture, and mouthfeel attributes. The films produced should be transparent and free of air bubbles for aesthetic appeal and stability considerations.

#### 2.2.1. Chitosan

Chitosan, a natural cationic polymer composed of a linear sequence of monomeric units linking 2-acetamido-2-deoxy-D-glucopyranose and 2-amino-2-deoxy-D-glucopyranose through β (1 → 4) glycosidic bonds, is soluble only in aqueous acidic solutions due to its crystalline nature [30]. Chitosan exhibits numerous advantageous characteristics, including its biodegradability, biocompatibility, minimal toxicity, antimicrobial capabilities, and, notably, its excellent mucoadhesive property, which is a crucial aspect of film formation [31,32]. To develop the appropriate ODFs, it is crucial to consider the molar mass, particle size, and concentration in the chitosan solution [33]. Based on the research conducted, it was reported that low-molecular-weight chitosan results in films with superior physical-chemical characteristics and more favorable sensory evaluations compared to medium-molecular-weight chitosan [34]. It is worth noting that all the films exhibited rapid disintegration; however, an increase in the chitosan concentration resulted in prolonged disintegration times. Furthermore, the incorporation of super-disintegrating agents, such as sodium starch glycolate and crospovidone, in the fabrication of ODFs resulted in faster disintegration [35]. Chitosan-based ODFs have been formulated using a range of plasticizers, including glycerol, sorbitol, PEG 400, and PEG 600.

Chitosan films displayed both impressive tensile strength and flexibility; however, the incorporation of super-disintegrating agents led to a reduction in their mechanical strength. The properties of the chitosan-based films were quite similar to those of CMC-based films from the same study [35]. Additionally, chitosan served as a mucoadhesive agent in enhancing drug permeability for HPMC-based films [36]. However, because of its widespread regulatory acceptance, low cost, and ease of manufacture, HPMC polymer was utilized extensively in pharmaceutical applications [37]. In a recent investigation, hyperbranched chitosan (HPCN)-based oral thin film and nanofiber (HCNF), as well as chitosan-blended hyperbranched polyester film and nanofiber (CNHPN), were developed with donepezil as a drug [34]. The thin film was fabricated using the solvent casting process, and nanofiber was formed via the electrospinning process. The in vitro experiments showed that the nanofiber (HCNF 2) disintegrated rapidly (within 15 s) and released approximately 97.03% of the drug within 45 min. The in vivo pharmacokinetic investigations revealed that the (CNHPN 2) formulation achieved a peak plasma concentration (Cmax) of 18.94 ng/mL after 3.3 h. Moreover, the total area under the curve for HPCN displayed a swifter absorption rate (978.1 ng/mL), surpassing that of the commercially available tablet formulation, which recorded 132.05 ng/mL. These results suggest that the chitosan-blended or HPCN nanofiber formulation might serve as a viable substitute for commercial formulations in the treatment of Alzheimer’s disease. Using electrospinning technology, aspirin-loaded fast-dissolving oral films composed of chitosan/pullulan composite nanofibers were created [31]. The solution properties and morphology of the resulting nanofibers were affected by the chitosan-to-pullulan ratio, particularly when there was an increase in the chitosan content. The scanning electron microscopy (SEM) results suggested that the diameter of nanofibers increased first and then decreased with an increased chitosan content in the solutions. Water solubility testing confirmed that the fast-dissolving oral films dissolved entirely in water within 60 s. In a different trial, frovatriptan-loaded ODF was fabricated using HPMC E15 as a film former, mannitol as a release modifier, glycerine as a plasticizer, and chitosan or NaCMC as the mucoadhesive polymer [38]. The drug was loaded using the optimized combination of excipients, together with 1–2% *w*/*w* of mucoadhesive polymers like chitosan or NaCMC, facilitating an enhanced interaction with the sublingual mucosa. The optimized film exhibited a variety of favorable physicochemical characteristics, including a reduced disintegration time (<2 min), a rapid dissolution rate (89.9% within 5 min), and good permeability. Furthermore, pharmacodynamic models provided additional evidence of the sublingual film’s effectiveness in migraine treatment.

#### 2.2.2. Pectin

Pectin is a naturally occurring anionic polysaccharide, primarily composed of poly α1–4-galacturonic acids. Its varying levels of methylation and amidation in its composition contribute to its diverse properties and applications in various fields [39]. Although pectin has found applications in various drug delivery systems, such as gel beads, films, and matrix tablets, there is a lack of information regarding its use as a film-forming material in ODF formulations [40]. To develop ODFs, researchers found that higher pectin concentrations resulted in thicker, more flexible films with longer disintegration times, suggesting the benefits of using lower pectin concentrations in pectin-based ODF development [41].

Aprepitant (APT), an antiemetic drug, was incorporated into an ODF using the solvent casting technique. The film-forming agent used was pullulan, and tamarind pectin was employed as the wetting agent, while liquid glucose served as both the plasticizer and solubilizer in this development process, as reported [42]. The optimized formulation displayed a uniform film surface, was non-sticky, disintegrated within 18 s, and demonstrated in vitro release with more than 87% of APT being released, which was higher than the release rates of Aprecap, micronized APT and non-micronized APT formulations. The animal preference investigation demonstrated a positive response to the film, while the in vivo pharmacokinetic study conducted on rabbits revealed that the relative bioavailability of the APT-loaded ODF was 1.80, 1.56, and 1.36 times greater than that of Aprecap, micronized APT, and non-micronized APT, respectively. This study suggests that the APT-loaded ODF has the potential to serve as an effective antiemetic option for cancer chemotherapy.

#### 2.2.3. Polyvinyl Alcohol (PVA)

PVA is a synthetic polymer that is semi-crystalline or linear in texture and can be granular or in powdered form. It is typically creamy or whitish, tasteless, odorless, non-toxic, biocompatible, and is able to withstand high temperatures [43]. PVA is offered in various grades in the market that vary based on the degree of hydrolysis and viscosity. By utilizing the solvent casting technique, the creation of ODFs based on PVA has effectively achieved the amorphization of the poorly soluble drug phenytoin, as reported [44]. Among the different formulations that were developed, the formulation labeled as PVA-S4, which consisted of 1% *w*/*w* PVA, 0.04% *w*/*w* sodium starch glycolate, and cosolvents like polyethylene glycol 400 (PEG 400), glycerin, and water, exhibited encouraging results, particularly in terms of a shorter disintegration time (1.44 min) and high drug content (100.27%) Moreover, the PVA-S4 film exhibited a higher dissolution rate compared to the film without a cosolvent, indicating the potential of cosolvents in enhancing the solubility and film dissolution of poorly water-soluble drugs. In summary, this study highlights the impact of cosolvents on the solubility enhancement of poorly water-soluble drugs and their film dissolution. The effectiveness of Soluplus^®^ (Ludwigshafen, Germany)/PVA-based oral thin films as a carrier for poorly soluble drugs like spironolactone in pediatrics has been demonstrated [45]. According to the findings, the most desirable formulation consisted of a 10% polymer concentration with a Soluplus^®^:PVA ratio of 0.33:0.66 and was plasticized with 30% PEG 400, which had a desirability value of 0.836. Freeze-dried film improved the in vitro dissolution of spironolactone in the optimized formulation by generating a more porous film compared to the non-freeze-dried version.

#### 2.2.4. Alginic Acid

Alginates are natural anionic polysaccharides that consist of β-D-mannuronic acid (M) and α-L-glucuronic acid (G) monomers linked together in (1 → 4) glycosidic bonds. The properties of the biopolymer depend on the composition and covalent bond of these monomers, made up of either similar or alternating (MMMM, GGGG, or GMGM) monomer blocks. Alginates are widely used in various applications, such as films, membranes, hydrogels, microparticles, beads, and tissue engineering applications. Alginate exhibits numerous favorable attributes, including biocompatibility and biodegradability, minimal toxicity, stability under physiological conditions, and the capacity to form gels when exposed to divalent cations such as Ca^2+^ and Mg^2+^. Even though sodium alginate films are highly hydrophilic, they are typically cross-linked with Ca^2+^ to improve their mechanical properties and water resistance. Alginate is additionally employed in the preparation of ODFs as a film-forming agent, either independently or in conjunction with other polymers [39]. A film-forming alginate hexyl amide derivative was prepared and used to enhance the bioavailability of repaglinide [46]. Sodium alginate was used as a film-forming polymer by utilizing many drugs, namely nebivolol [47], cetirizine [48], and piroxicam [49]. The disintegration time of the sildenafil-loaded PVA/sodium alginate-based ODF was demonstrated to be reduced with an increasing concentration of sodium alginate in the formulation [50]. ODFs based on alginate exhibit an average tensile strength and high hydrophilicity, and they disintegrate within 60 s. In contrast, when comparing films made from HPMC to HPMC combined with L-arginine/poloxamer as a solubilization system, the latter showcased increased mechanical strength while maintaining similar disintegration times [49].

#### 2.2.5. Hydrolysed Collagen and Gelatin

Gelatin is a protein derived from collagen through thermal denaturation and partial hydrolysis in either acidic or alkaline environments. Its structure consists of a blend of α-, β-, and γ-chains, with a typical amino acid composition that includes Ala-Gly-Pro-Arg-Gly-Glu-4Hyp-Gly-Pro. The gelatin in aqueous solutions can create films and gels, showcasing characteristics such as adhesion, cohesion, transparency, thickening, and water-retention capacity. Gelatin finds applications in wound dressings, drug delivery systems, and oral films, owing to its attribution of biodegradability, biocompatibility, non-immunogenicity, mucoadhesiveness, and film-forming ability [51]. Incorporating low-molecular-weight hydrolyzable collagen into gelatin-based ODFs improves the film’s flexibility, increases its hydrophilicity, and leads to rapid disintegration [52]. Several studies have shown that gelatin-based ODFs outperform other natural polymers in terms of tensile strength and mucoadhesiveness [53,54]. However, when gelatin was utilized alone, the films exhibited poor hydrophilicity. To enhance the hydrophilicity and reduce the disintegration time of the ODFs, gelatin was blended with various substances such as hydrolyzed collagen, starch, CMC, and HPMC [53,54,55]. Increasing the proportion of these blend components led to increased hydrophilicity and a reduced disintegration time for the ODFs. 

In a recent study, the characteristics of ODFs fabricated with pregelatinized starch, gelatin, and acerola powder, specifically investigating the impact of macromolecule concentrations, were described [53]. The ODFs were free of insoluble particles; however, an increasing starch concentration led to greater surface heterogeneity and roughness, observed through atomic force microscopy. The inclusion of starch also increased the hydrophilicity, resulting in improved disintegration times in both the in vitro and in vivo analyses. Fourier-transform infrared spectroscopy demonstrated a favorable interaction between carbohydrates and proteins in the ODFs, and even following exposure to severe conditions for 50 days, the ODFs preserved a minimum of 60% of their antioxidant capacity. Hence, the ODFs with acerola powder hold great potential as a delivery system for active compounds. The films maintained their mucoadhesiveness at a constant level, regardless of the proportion of starch or gelatin [54]. However, with HPMC increasing the proportion of gelatin in the blends, a higher mucoadhesiveness resulted [56].

It has been proposed that the source of gelatin utilized can have a substantial impact on the properties of ODFs [57]. While there is a scarcity of research on collagen’s application in the formulation of ODFs, it appears that hydroxyl collagen can enhance both the hydrophilicity and disintegration characteristics when blended with other biopolymers. 

#### 2.2.6. Pullulan

Pullulan is a neutral, naturally occurring polysaccharide extracted from the fermentation media of *Aureobasidium pullulans*. Notably, it possesses low oxygen permeability and exhibits excellent film-forming and mechanical characteristics, in addition to being transparent, flexible, non-toxic, biocompatible, and biodegradable. Pullulan-based ODFs containing amlodipine besylate were developed using the solvent casting method, which demonstrated acceptable mechanical properties, disintegration times, dissolution, and toxicity [58]. The development and characterization of iron-loaded pullulan-based ODFs have been reported [59]. The bitter taste of iron was masked using a microencapsulation technique, and the films were fabricated employing a modified solvent casting method. It was concluded that the film’s former pullulan can be successfully scaled up for commercial use in various nutritional supplements and active ingredients. Pullulan can be considered a promising film-forming agent for ODF development, especially for pediatric patients. 

A novel approach to using ODFs was assessed as an oral vaccination delivery system [60]. To ensure the antigenicity of ODFs, trehalose was utilized as a stabilizer, while pullulan was employed as a film-forming agent. The optimal ratio of sugar and polymer was determined based on the model antigen, β-galactosidase. Subsequently, ODFs containing the whole inactivated influenza virus vaccine (WIV), H5N1, were formulated, and the stability of the H5N1 hemagglutinin, a crucial component, was estimated by measuring hemagglutination activity. In summary, a variety of trehalose and pullulan compositions effectively maintained the stability of both β-galactosidase and the WIV within ODFs. As a result, ODFs emerge as a promising approach for delivering WIV via the oral cavity, potentially providing a practical alternative to conventional injections. In addition, pullulan-based ODFs have found application as class II drug carriers combined with xanthan gum as a thickening agent and glycerol as a plasticizer, demonstrating remarkable uniformity in drug content, high tensile strength, and minimal elongation at the point of fracture. These findings suggest that pullulan serves as an effective polymer matrix for enhancing the rapid release of poorly water-soluble drugs, ultimately enhancing their bioavailability [61].

#### 2.2.7. Starch

Starch is an abundant natural polysaccharide comprised of two macromolecules: amylose and amylopectin. Amylose is a linear polymer of the α-1,4 anhydroglucose units that form a colloidal dispersion in hot water and exhibit excellent film-forming ability. In contrast, amylopectin is a highly branched polymer of the α-1,4 anhydroglucose chains linked by α-1,6 glucosidic branching points and is entirely insoluble [39]. The semi-crystalline characteristics of native starch can present unfavorable properties, like poor aqueous solubility or decreased mechanical strength. It may be chemically, enzymatically, or physically modified to enhance its properties and functionality [62]. In addition to their transparent, odorless, tasteless, and biodegradable nature, starch materials possess favorable properties for developing ODFs [51]. Various researchers have utilized starch in ODF formulations, either as the main film-forming agent [63], in combination with other biopolymers [54], or blended with synthetic or semi-synthetic polymers [64]. In the investigation of wheat starch blended with HPMC and PEG for the development of ODFs through freeze-drying and heat-drying methods, it was found that higher concentrations of HPMC and starch resulted in an increased tensile strength and disintegration time [65]. The optimized proportion of components (Starch: HPMC: PEG, 3:1:3) demonstrated a lower tensile strength but better folding endurance and disintegration time compared to the heat-dried films, indicating the influence of the production method on film properties. Starches, particularly modified or pregelatinized starches, are highly suitable as film-forming agents for ODFs. They have the ability to form homogeneous and hydrophilic films and exhibit good mechanical properties, rapid disintegration, and strong mucoadhesiveness [66].

Modified starches serve as effective film formers for ODFs, offering improved properties such as enhanced flexibility, increased mechanical strength, rapid disintegration, and dissolution. Recently, physically modified rice starch was produced through both alcoholic-alkaline treatment and planetary ball milling [63], resulting in a reduction in crystallinity, an increase in swelling capacity, and enhanced cold water solubility compared to native starches. Desloratadine-loaded ODFs, created using the solvent casting method, demonstrated superior flexibility and rapid disintegration (62.3 ± 23.1 s), achieving over 95% dissolution in under 10 min. The corn starch, gelatinized and plasticized through microwave processing, demonstrated satisfactory mechanical and physical attributes, affirming its potential as an effective film former [67]. This was further confirmed by incorporating the model drug, diclofenac sodium. Modified starches like Lycoat RS 720 (pregelatinized hydroxypropyl pea starch) can often function as disintegrating agents in ODFs. An optimized formulation of racecadotril ODFs successfully achieves favorable mechanical properties, disintegrating rapidly within a few seconds, and releasing over 85% of the drug within 5 min across various dissolution media [68].

#### 2.2.8. Maltodextrins (MDXs)

MDX, which is typically derived from the hydrolysis of starch, consists of three to nineteen units of D-glucose molecules, connected mainly through α-(1 → 4) glycosidic bonds and with infrequent α-(1–6) branches. It is categorized based on its dextrose equivalent, which ranges from 3 to 20. MDX possesses several advantageous properties, including its effectiveness as a film former, lack of odor, high solubility, low hygroscopic nature, excellent carrier function, non-toxicity, biodegradability, and biocompatibility. While MDX easily dissolves in water, it exhibits limited solubility in anhydrous alcohol [69]. The modified solvent casting method was employed to produce benzydamine-loaded ODFs using two types of MDX, which were plasticized with sugar alcohols like xylitol and sorbitol, along with the super disintegrant Kollidon^®^ (Ludwigshafen, Germany) CL-F [70]. Films that were plasticized with xylitol exhibited a quicker disintegration time (ranging from 17.6 ± 2.9 to 29.2 ± 3.8 s) compared to the films containing sorbitol (with disintegration times between 23.8 ± 2.9 and 31.7 ± 3.9 s). The addition of the super disintegrant Kollidon^®^ CL-F did not significantly impact the disintegration time. A patent application titled Self-supporting Film for Pharmaceutical and Food Use (EP1689347) introduces an ODF that exclusively utilizes MDX as the film-forming material [71]. According to this patented method, MDXs make up 40–80% of the ODF composition, and these films disintegrate within 1 min, leaving no residues in the mouth, in contrast to hydrocolloids, which tend to leave a residue behind. A recently introduced ODF containing sildenafil citrate, composed of MDX, is now available on the market. These ODFs come in four different dosage forms for the first time, including 25, 50, 75, and 100 mg. The bioequivalence test demonstrated satisfaction for sildenafil and N-desmethyl-sildenafil in terms of both the rate and extent of bioavailability [72].

Clinical trial findings indicate the disclosure that the bioavailability of externally administered calcifediol (Vitamin D3), when given in the form of a 25,000 IU ODF, is comparable to or even superior to a similar dose of the standard oral solution when administered under the same (fed) conditions, in terms of both the rate of absorption (Cmax) and the extent of absorption within a specified time frame (AUC_0–t_ = 72 h) [73]. Additionally, the ODF comprising 25,000 IU of vitamin D3 was found to have a mild and enjoyable taste, with a slight aftertaste. Overall, the ODF presented a valuable substitute for the commercially available oral solution. In recently published research, an MDX-based ODF loaded with 30 mg of iron as pyrophosphate and 400 µg of folic acid was successfully developed and scaled up. A crossover clinical trial compared the optimized ODF with a highly bioavailable Sucrosomial^®^ (London, UK) iron capsule. Pharmacokinetic data indicated that the rate and extent of elemental iron absorption from the ODF were comparable to those achieved with the standard iron capsule, establishing it as a suitable product for oral iron supplementation [74].

One CQA for ODFs involves developing dosage forms with appropriate tensile properties for packaging, transportation, and patient handling. The addition of an amorphous, water-insoluble nanofiller, specifically polyvinyl acetate, significantly enhanced the tensile strength (1.5-fold) and the elastic modulus (4-fold) of MDX-based ODFs [75]. Analyses through DSC and ATR-FTIR revealed the effectiveness of polyvinyl acetate nanoparticles as a reinforcing agent, particularly at concentrations of 3% and 5% *w*/*w*.

### 2.3. Plasticizers

Plasticizers are low-molecular-weight additives added to a polymer solution to promote plasticity and flexibility. Plasticizers lower the glass-transition temperature of the polymers from a hard, glassy material to a soft, rubbery material. It is important to highlight that the moisture absorption of various plasticizers plays a crucial role in influencing various film characteristics [76]. Lipid plasticizers, such as fatty acids and their derivatives, lecithin, oils, and waxes, can be used to decrease the water vapor permeability of films due to their non-polar or hydrophobic nature [77]. However, the inclusion of lipids can introduce gloss and improve the visual appeal, yet it might impact the film’s cohesiveness and structural integrity, including a potential reduction in film strength. 

Polysaccharide-based films possess a degree of rigidity, which necessitates the use of plasticizers to enhance their pliability for easier handling. Among the different plasticizers, glycerine can be considered an effective plasticizer, as seen with starch, methylcellulose, highly carboxymethylated starch, and HPMC films [78]. At equivalent concentration levels, hydrophilic plasticizers with smaller molecular weights, such as propylene glycol and glycerine, resulted in higher water vapor permeability compared to larger molecular weight plasticizers like PEG 400 in methylcellulose films. The effect of different plasticizers (glycerol, vitamin E TPGS, and triacetin) and their concentrations on the physicomechanical properties of pullulan-based oral films was disclosed [79]. Typically, the flexibility of protein-based films such as casein, gelatin, collagen, zein, and soy protein are commonly enhanced with plasticizers like glycerine, propylene glycol, sorbitol, PEG, sucrose, and oleic acid [80]. Apart from the cost, when choosing a plasticizer, it is essential to take into account three fundamental factors: compatibility, effectiveness, and durability. 

The addition of plasticizers at a concentration ranging from 0–20% by weight of the dry polymer is employed to prevent undesirable physicomechanical and thermal properties of the polymeric films [79]. Hydrophilic cellulose-based polymers can be effectively combined with plasticizers containing polar hydroxyl groups, such as PEG, propylene glycol, glycerine, and polyols. In contrast, less hydrophilic cellulosic polymers can be plasticized with esters of citric acid and phthalic acid. Glycerine is frequently utilized to plasticize PVA due to its small molecular weight, low volatility, and compatibility with the polymer matrix [81]. Hydrophilic compounds, such as polyols (glycerine and sorbitol), are commonly used in starch films [82]. It is important to consider the characteristics of plasticizers to lower the glass-transition temperature of polymers to between 40 and 60 °C for non-aqueous solvent systems and below 75 °C for water-based solvent systems. Diethylene glycol can be used as a plasticizer for both HPMC- and PVA-based polymer films [83]. It was reported that the water absorption of Eudragit films is influenced by the specific Eudragit polymers and plasticizers employed. For instance, Eudragit E films, plasticized with diethyl phthalate, dibutyl phthalate, and tributyl citrate, exhibited a higher level of water absorption compared to those plasticized with triacetin [84]. A study examining the impact of varying PEG molecular weights and concentrations on the mechanical and thermomechanical attributes of free HPMC films revealed that the inclusion of a plasticizer led to a reduction in both of these characteristics [85].

The nature and quantity of plasticizers can indeed affect drug release by reducing the secondary bonding between polymer chains, thereby improving the mobility of the drug. Small amounts of plasticizer may make the polymer more rigid instead of making it softer, which is referred to as antiplasticization. This phenomenon can be leveraged as a formulation technique to control the permeability of drugs in the polymeric systems used in pharmaceuticals [86]. The release of the drug from a polymer system with plasticizers primarily depends on the physicochemical characteristics, specifically the solubility parameters of the plasticizer and the extent of plasticizer leaching. Plasticizers with lipophilic properties, such as dibutyl sebacate, typically remain within the polymeric system, resulting in the formation of robust and mechanically resilient coatings as the drug is released. Nonetheless, hydrophilic plasticizers have a tendency to wash out, which can result in reduced mechanical strength, potential cracking, or an increase in pore formation.

### 2.4. Sweetening and Flavoring Agents

Various techniques are utilized to mask the bitter or unpalatable taste of drugs in oral thin films. These methods include the utilization of sweetening agents, ion exchange resins, microparticles, inclusion complexes, and nanocarriers [87]. In recent times, lipids have garnered significant attention from researchers for their role in taste masking [88]. Various techniques, such as hot-melt extrusion (HMA), melt granulation, spray drying/congealing, and emulsification, can harness the taste-masking properties by utilizing lipids effectively. Sweeteners are generally added to mask the unpleasant taste and odors of APIs or excipients, improving palatability and patient acceptability. Sweeteners used in ODFs can be mainly classified into two types: natural sweeteners and artificial sweeteners. Glycyrrhizinic acid, derived from the root and rhizome extracts of licorice (Glycyrrhiza glabra), is a triterpenoid saponin widely utilized as a natural sweetener, which has been reported to be at least 30 times greater than sucrose [89]. Rebiana, a natural sweetener derived from the plant Stevia, possesses sweetness more than 200–300 times that of sucrose [90]. Despite artificial sweeteners being significantly sweeter than natural sugars, they are often associated with an aftertaste effect that can be alleviated by combining natural and artificial sweeteners. Typically, sweeteners are employed in concentrations of 3 to 6% *w*/*w*, either individually or in combination. Mannitol, a polyhydric alcohol, is extensively utilized as a sugar alcohol sweetener in ODFs due to its ability to enhance the physical integrity of the dosage form [91]. Additionally, mannitol offers exceptional stability and compatibility with APIs and has a non-hygroscopic nature, which is advantageous for ensuring the long-term integrity and stability of the dosage form. Sweeteners from significant sources, including sucrose, dextrose, fructose, glucose, liquid glucose, and maltose, are limited in their application within the diabetic community due to their substantial caloric content [92].Combining polyhydric alcohols like sorbitol, mannitol, isomalt, and maltitol can offer a pleasant mouthfeel and a refreshing cooling sensation. Aspartame is a dipeptide that provides a sugar-like taste at low concentrations and is nearly 200 times sweeter than sucrose. Aspartame has high stability, good compatibility with other excipients, and low caloric value, making it a preferred choice for low-calorie or sugar-free formulations [93]. Even though aspartame is used in the food and pharmaceutical sectors, a thorough understanding of its pros and cons is essential, particularly concerning potential health risks in children [94].

Sucralose, a chlorinated derivative of sucrose, exhibits excellent stability and compatibility with other ingredients over a wide pH range and is recognized for its absence of adverse effects on dental health. With approximately 200 times the sweetness of sucrose, acesulfame potassium provides a quick onset of sweet taste. Notably heat stable, pH stable, and non-reactive in Maillard reactions, acesulfame potassium proves suitable for diverse formulation and manufacturing processes. These sweetening agents, along with appropriate flavoring agents, play a critical role in improving the taste, palatability, and patient compliance of ODFs and ODTs. The selection of a sweetener depends on factors such as the taste profile, stability, compatibility, manufacturing process, and the target patient population.

The extensive contact between oral films and the oral mucosa underscores the significance of directing a part of the development efforts toward creating a pleasing and palatable formulation. Perception of the flavors depends on the ethnicity, age, and liking of the people [95]. In ODF formulations, it is common to add flavors at a concentration of up to 10% *w*/*w*. To enhance the flavor intensity and improve the mouthfeel attributes, cooling agents can also be introduced. Flavoring agents play a crucial role in enhancing the palatability and overall sensory experience of ODFs. The selection and incorporation of appropriate flavoring agents are essential to mask any unpleasant taste or medicinal bitterness that may be associated with the API or other excipients in the film. Flavoring agents not only improve the taste but also contribute to patient compliance, especially in pediatric and geriatric populations.

### 2.5. Saliva-Stimulating Agent

Saliva-stimulating agents, also known as sialagogues, are substances that can increase saliva production in the mouth. They can be beneficial for individuals with conditions such as dry mouth (xerostomia) or for those who may have difficulty producing adequate saliva [14]. The inclusion of these agents can expedite the disintegration process, leading to the rapid dissolution of ODF within a short time. Frequently employed agents for stimulating salivation in rapidly dissolving strip formulations encompass acids like citric acid, malic acid, lactic acid, ascorbic acid, and tartaric acid. These substances are usually included individually or in combination, with concentrations typically falling in the range of 3 to 5% by weight relative to the film. Furthermore, xylitol, a type of sweetener, has been proven to possess saliva-stimulating properties, as evidenced by studies conducted on chewing gum [96]. 

### 2.6. Coloring Agents

Coloring agents are often added to ODFs to enhance their appearance or differentiate between different formulations. Specific coloring agents can vary depending on the regulatory requirements, formulation considerations, and desired aesthetics. Commonly used coloring agents in ODFs include pigments such as titanium dioxide to achieve a lighter or opaque appearance in the film and iron oxides to provide shades of red, yellow, or brown in the film. Moreover, FD and C-approved coloring agents, such as indigo carmine and brilliant blue dyes, are used to provide a blue color, fast green FCF (for coloring food) to create green, and tartrazine dye is often used to achieve a yellow color in the film [58]. Agents should be incorporated into the ODF formulation at a maximum concentration of 1% by weight, specifically when certain ingredients or medications within the formulation are insoluble or in suspension [97]. It is important to note that the use of coloring agents in pharmaceutical products is subject to regulations and guidelines set by regulatory authorities, such as the U.S. FDA or the European Medicines Agency. These agencies provide specifications and limitations on the use of coloring agents to ensure safety and suitability for oral consumption.

### 2.7. Stabilizing Agents

These are essential components in ODFs to maintain their structural integrity, prevent degradation, and ensure the desired properties of the film [49]. These agents help in preserving the film’s texture, flexibility, and ability to dissolve quickly. Several stabilizing agents are commonly used in ODF formulations, including HPC, PVP, xanthan gum, sodium alginate, locust bean gum, carrageenan, gelatin, sorbitol, and surfactant [98]. Typically, these agents enhance the viscosity and uniformity of the dispersion, collaborating with other excipients and active ingredients to establish a unified and enduring ODF dosage form. In order to enhance the film properties, such as spreadability and dispersibility, generally, minor quantities of surfactants and emulsifying agents are included. The specific choice and combination of stabilizing agents depend on the desired characteristics of the film and the specific requirements of the formulation.

## 3. Manufacturing Methods

The primary methods utilized to prepare ODFs include the solvent casting technique, electrospinning, hot-melt extrusion (HME), and the emerging technique of 3D printing (3DP) for personalized dosing [99]. When developing a film formulation and its processes, it is beneficial to consider the drug product CQAs that align with the quality target profile attributes.

### 3.1. Solvent Casting

At present, solvent casting has emerged as the widely preferred technique for practical, feasible manufacturing and the scaling up of oral films [100]. On a small scale, for the preparation of ODFs in a laboratory setting, polymer(s) and other necessary ingredients are dissolved in a suitable solvent to form a homogeneous solution. The solution is then cast onto a substrate, and the solvent is allowed to evaporate, leaving behind a thin film. The manufacturing process of ODFs on an industrial scale using the solvent casting technique consists of multiple steps, starting with the precise dispensing of the drug, excipients, and non-toxic FDA-approved class III solvents, followed by their homogeneous dispersion in a low-shear or high-shear mixer under thermostatic control [4]. Nonetheless, it is not recommended to utilize high-shear mixers when dealing with encapsulated drug actives, as this method can lead to the removal of the encapsulating material. The slurry undergoes solvent evaporation within a hot-air oven set at a designated temperature. Subsequently, it is applied onto a meticulously chosen liner using a knife-over-roll coater equipped with a precise pin gauge. The resulting dried laminates are rolled up into master rolls and subsequently cut into individual dose units based on the desired dimensions of the ODF. These single-dose units are enclosed within pouches or sachets using packaging and sealing machinery. The solvent casting method used to prepare ODFs is similar to the manufacturing process of buccal film, as depicted in Figure 1. The child-resistant and tamper-proof packaging material is designed to protect against adverse environmental conditions and ensure ease of use. It is of utmost importance to determine the weight of each film unit being packaged, as the drug dosage in the ODF is directly dependent on its weight. One of the key advantages of this dosage form is its flexibility in generating multiple dose units simply by adjusting the size of the film. Various factors, such as the solvent evaporation rate, airflow velocity of the hot air, pin gauge dimensions, and conveyor belt speed, influence the formation of the cast film. Scaling up the production of ODFs using the solvent casting technique poses several challenges. Equipment adaptation is often necessary, and the transition to larger-scale equipment must preserve the desired film characteristics. Maintaining uniformity and homogeneity of the film on a larger scale is crucial, as variabilities in the mixing and casting processes can lead to inconsistencies. Ensuring the integrity of the film, and avoiding defects such as cracking or uneven thickness is a challenge that grows with the scale of production. Implementing effective quality control measures becomes more intricate, demanding rigorous batch-to-batch consistency and adherence to regulatory standards. The increased scale also introduces considerations of elevated costs related to equipment, materials, and energy consumption. Environmental factors like temperature and humidity become more pronounced in larger-scale production, affecting the drying process and the overall quality of the films. Addressing these challenges requires meticulous planning, optimization of processes, and a commitment to maintaining high-quality standards throughout the scaled-up production of ODFs.

Quetiapine fumarate ODF production, at a pilot scale, has been executed through the solvent casting method, following the principles of quality by design [101]. The optimized films were prepared by employing HPMC E5 as the film former and propylene glycol as the plasticizer at a temperature range of 65–70 °C.

The viscoelastic characteristics of the casting solution or dispersion significantly influence the film’s characteristics, such as content uniformity, thickness, morphology, and drug release. To overcome the typical problem of poor drug loading, investigators have developed a porous ODF using HPMC polymer via the solvent casting method [10]. A proposed solution for addressing the issue of content uniformity is the introduction of a new unit-dose plate that can be filled with the required volume of a constituent mixture [102]. An alternative method to ensure the content’s uniformity is by using doctor-blade film coaters, wherein a slurry is uniformly applied to a substrate through a metering blade to achieve the desired thickness [100]. To ensure the uniformity of the dispersion, the rheological properties and solids’ content are estimated, and in-process sterility testing is conducted to identify any potential bioburden. Various researchers have observed that the mechanical properties of the films, such as folding endurance, tensile strength, toughness, puncture strength, and film thickness, were significantly impacted by the formulation compositions [70,75]. For instance, different domperidone ODFs prepared using PVPK-90 were discovered to significantly influence the physical properties, mechanical properties, film thickness, and drug release rate of the resulting films [103]. The formation of the cast film on the selected liner is influenced by several variables, which encompass the rate of solvent evaporation, air flow velocity, positioning of the heat source, pin gauge dimensions, and conveyor belt speed [104]. The characteristics of the intermediate liner, such as the contact angles and surface tensions, can influence the film’s quality. Investigations conducted on hydrochlorothiazide ODFs prepared with HPC or HPMC as film-forming agents have demonstrated remarkable influences of the production process on the film’s attributes [105]. The film can be manufactured similarly as a transdermal patch—as a large sheet that can be cut into specific dimensions consisting of individual dosage units and packaged in pharmaceutically acceptable packaging materials. Such a formulation advancement would enable users to readily recognize their medication, thereby enhancing safety and adherence. Lately, researchers have demonstrated that the inclusion of nanocrystal dispersions or microparticles in ODFs can lead to improved drug dissolution rates [106] or enable extended drug delivery [107]. Nevertheless, it is important to consider that the inclusion of such particles may potentially impact the mechanical characteristics of the film.

### 3.2. Three-Dimensional Printing (3DP) Method

The utilization of additive manufacturing techniques such as 3DP has significantly reduced the preparation time while improving the mechanical characteristics of films. Numerous 3DP methods, including the fused deposition method (FDM), HME, and print-fill, have been extensively studied, with thermoplastic polymers serving as the primary components [108]. Among these methods, FDM is particularly well-suited for special populations requiring customized dosing or controlled release kinetics. The feasibility of using HME 3DP for preparing ODFs with MDX was investigated [108]. To determine the printable design space for individualized formulations, the impact of the critical formulation and process variables were evaluated. These personalized films have the potential to be employed in a clinical setting, maximizing therapeutic efficacy and minimizing adverse drug reactions. In 2015, the U.S. FDA approved SPRITAM^®^ (levetiracetam) (Langhorne, PA, USA), marking a milestone as the first ODT formulation with a porous structure manufactured through 3D printing using the Zip^®^Dose technology. Because 3D printing can produce dosage form designs with performance characteristics difficult to create with conventional methods, personalized ODFs also have the potential to improve drug efficacy and/or reduce adverse events, which should lead to improved patient outcomes [109,110]. As part of a study, a thermal inkjet printer designed for commercial use was adapted to print personalized ODFs that incorporated varying doses (1.25 and 3 mg) of warfarin. The fabrication of the film involved the use of HPMC (20% *w*/*w*) and glycerol (3% *w*/*w*) as its constituent materials [111]. To address the challenge of taste masking in ODFs, the production of multilayered films with separate taste-masking layers using FDM 3DP has been outlined [111]. Filaments were created by blending PEO and PVA with ibuprofen and acetaminophen as model drugs, each at designated temperature ranges. In contrast to the 3DP methods previously documented, the oral adhesive films fabricated through FDM exhibited superior structural integrity and content uniformity. The FDM technique faces a significant limitation due to its elevated processing temperature, which greatly restricts its practical utility for thermolabile pharmaceuticals. It was proposed that a combination of FDM and inkjet printing can be efficiently utilized to incorporate thermolabile drugs [112]. The utilization of extrusion 3DP, which directly extrudes and prints semisolid materials like gels or pastes (Figure 2), offers a solution to overcome the limitation of the FDM process, specifically the need for material filaments [113]. 

### 3.3. Hot-Melt Extrusion (HME)

HME technology is a solvent-free, single-step continuous manufacturing process commonly employed to improve the water solubility of hydrophobic drugs through the formation of solid dispersions comprising both drugs and carrier materials (Figure 3). HME offers a promising and practical alternative to solvent casting, as it minimizes solvent usage and addresses issues associated with mixing and drying processes. An innovative approach combining the HME technique and solvent casting method has been utilized to develop ODFs incorporating micropellets, enabling extended drug release [107]. Manufacturing of a non-sticky, transparent, and uniform film using a hot-melt extrude equipped with a standard screw configuration has been described. The polymer matrix used for the melt extrusion process was comprised of modified starch with glycerol in addition to sweetening and saliva-simulating agents. The film formulations demonstrated fast disintegration (6 to 11 s) and demonstrated over 95% dissolution within 5 min [27]. In a recent investigation, HME was employed to achieve the continuous production of ODFs containing a substantial dosage of nimesulide, classified as a BCS II compound [114]. A mechanically robust film was successfully formulated, exhibiting an exceptional drug-loading percentage of up to 21.73% *w*/*w* and showcasing an optimal drug-release profile, further highlighting its effectiveness. The use of HME technology effectively improved the taste and enhanced the aqueous solubility of the drug in the development of mefenamic acid-loaded ODTs. This approach has the potential to be applied to the development of ODFs embedded with taste masking of bitter APIs, especially those belonging to BCS class II and class IV drugs [115].

ODFs containing acetaminophen have been produced using Klucel™, HPC E5, and Soluplus^®^ as carrier materials. The advanced physicochemical analyses, including Raman mapping, revealed the existence of drugs in an amorphous state in both the HME and 3D-printed films. Nevertheless, a relatively consistent distribution of amorphous drugs was observed in the 3D-printed films in comparison to the films fabricated using other techniques [116]. The findings indicate that combining the HME and 3DP methods holds significant promise for improving the physical attributes of formulations and producing ODFs with favorable features, such as rapid drug dissolution rates.

An extended drug-release film created by incorporating matrix drug-loaded particles has been reported [117]. These microparticles were fabricated by employing HME followed by milling, with diclofenac used as the model drug and Eudragit^®^ RS (Essen, Germany) serving as the matrix former. The ODF was subsequently made by uniformly dispersing the microparticles in the ODF-forming polymer, which consisted of 15% HPMC and 6% glycerol as the plasticizer. Although the process faced challenges related to sedimentation and agglomeration, the incorporation of microparticles with a size of up to 500 µm resulted in ODFs with even drug distribution and satisfactory physical and mechanical properties. 

### 3.4. Electrostatic Spray Deposition

The electrostatic spray process, also known as electrostatic spray deposition (Figure 4), applies a charged coating material, typically in the form of powders or liquid droplets or film, onto a charged surface using an electrostatically charged spray [2]. The electrostatic attraction between the charged particles and the surface causes them to adhere and form a uniform coating. This technique is commonly used in various industries for applications such as painting, powder coating, and surface treatment. In dry powder coating, the process of film formation occurs through the merging and smoothing out of individual powder particles, facilitated by the application of heat. It is advantageous to use thermoplastic polymers or pre-plasticized coating compositions with low glass-transition temperatures, ideally falling within a temperature range between 40 and 60 °C. This allows for film curing to take place at relatively low temperatures, which is beneficial when dealing with thermally sensitive drugs [118]. The extent of plasticization or reduction in glass-transition temperature significantly influences the particles’ ability to coalesce during the film formation process.

A technique known as electrostatic powder deposition has been developed to create fast-dissolving PEO films for drug delivery, as detailed in the literature [119]. These films were prepared in various ways, including through a physical mixture of PEO and acetaminophen, as well as co-processed PEO and drug particles. The active films exhibited an average drug content of 97%, whereas the physical mixture of powders displayed more variability, with a relative standard deviation of 11.9%, compared to 1.8% in the films prepared using co-processed particles. Mechanical testing of the prepared films revealed that the active films could stretch up to 15%, in contrast to the Listerine^®^ (New Brunswick, NJ, USA) strips at 1.6% and PEO films at 3.8%, mainly due to the plasticizing influence of the drug on PEO in the drug-containing films. Both active films achieved more than 85% drug release within a short time (2 min), indicating the potential of electrostatic powder deposition for creating free films for drug delivery. Moreover, the study highlighted that the efficiency of powder deposition through spraying was significantly influenced by various processing factors, such as the charging voltage, the distance between the spray gun tip and the substrate, and environmental humidity, as reported [120]. Additionally, for effective powder deposition, the sprayed powders must possess optimal electrical conductivity, which can be improved by increasing the environmental humidity. Overall, the full potential of this technique is not fully explored and needs further investigation.

### 3.5. Electrospinning Method

The electrospinning method involves the transformation of polymer solution droplets from a spherical shape to a conical shape, resulting in the production of nanosized fiber filaments (Figure 5). Electrospun nanofibers have attracted considerable interest because of their distinctive characteristics and potential applications. These nanofibers feature a high surface area-to-volume ratio and impressive mechanical attributes and can be customized to exhibit specific functionalities, including controlled release, filtration, tissue engineering, and applications related to energy storage [121]. Optimizing the diameter and structure of the resulting fibers can be achieved by manipulating different factors, including the polymer concentration, solution viscosity, electric field intensity, and spinning distance. Electrospun oral films offer versatile applications, including tailored solutions for addressing specific oral conditions like periodontitis or mucositis, as well as serving as effective drug delivery platforms for systemic treatments. Electrospinning offers versatility in the design and fabrication of oral films, allowing for customization of their properties and characteristics [122]. This enables the development of personalized medicine approaches or targeted therapies tailored to specific patient needs. Furthermore, oral films composed of nanofibers demonstrate impressive mechanical robustness and flexibility, all the while enabling the precise and controlled release of medications at a predetermined rate [123]. According to the report, the hyperbranched nanofiber formulations displayed remarkable mechanical durability and effective drug delivery properties. These findings indicate that the hyperbranched cellulose nanofiber holds significant promise as a feasible alternative to commercially accessible dosage forms [124]. Sequential electrospinning was employed to successfully fabricate a multilayer film, with the exterior layers comprising ethyl cellulose nanofibers, while the inner layer contained gelatin nanofibers loaded with curcumin [125]. Such a strategy could be adopted to develop ODFs with incompatible drugs as a combination dosage form. A study compared electrospun orally disintegrating films with conventional casting films for the formulation of rizatriptan using PVP and PVA as polymers [126]. Electron microscopy imaging revealed that the electrospun-based films exhibited a nanoporous structure, whereas the casted films did not display any discernible pores. In both film types, the drug was evenly distributed, with no evident interactions between the drug and the polymers. Furthermore, the electrospun-based films demonstrated increased bioavailability and a faster Tmax in comparison with the marketed films and tablets. This investigation inferred that electrospun-based films outperformed the conventional casted films in terms of their in vivo pharmacological efficacy, potentially due to the nanostructure achieved via the electrospinning approach. Ensuring appropriate dosage forms for children is crucial for promoting treatment adherence and effective pharmacotherapy, particularly in low-resource settings where long-term treatment is required. A study has been conducted for the development of an isoniazid-loaded ODF based on an electrospinning technique utilizing a combination of natural and synthetic polymers [127]. The ODFs consisted of nanofibers with satisfactory thermal stability and potential drug amorphization. Rapid disintegration (<15 s) and rapid drug release of isoniazid in less than 60 s were noticed. The developed ODFs exhibited several features that are valuable for pediatric patients, including ease of administration, favorable drug loading, and rapid drug-release properties.

A feasibility study explored the use of the electrospinning process to create nanofibers loaded with sildenafil, employing pullulan as the film-forming polymer and water as the sole solvent [128]. By optimizing the electrospinning parameters, such as the shear force, viscosity, and surface tension of the polymer solution, a continuous production of nanofiber-based ODFs was achieved. It was demonstrated that the disintegration of electrospun products proved to be exceptionally rapid and compliant with both Ph. Eur., and USP standards. The delivery of biopharmaceuticals to the oral mucosa presents various promising applications in the field of medicine, such as localized treatment, enhanced patient convenience, and improved patient compliance compared to traditional injection-based delivery methods. The delivery of biopharmaceuticals through the oral mucosa presents several challenges, including the mucosal membrane barrier, enzymatic degradation, and achieving controlled release, which needs to be addressed for successful implementation. To overcome this challenge, researchers explored the use of a dual-layer mucoadhesive patch produced through electrospinning using lysozyme as a model antimicrobial protein [129]. Nanofibers were made of PVA/Eudragit RS100 polymers using an ethanol/water mixture as the solvent. The resulting fibrous membranes exhibited a cumulative drug release of 90 ± 13% within 2 h. Confocal microscopy along with dual fluorescent fiber labeling, demonstrated the even distribution of lysozyme and polymers throughout the patch. The formulation achieved high encapsulation efficiency (93.4 ± 7.0%) and preserved enzyme activity (96.1 ± 3.3%). This study underscores the considerable potential of this drug delivery system for delivering therapeutic proteins to the oral mucosa. While the electrospinning process has proven to be a valuable technique for fabricating nanofibers with a wide range of applications, it does have some limitations. There are a few primary limitations of the electrospinning process, including process complexity, limited scalability, solvent compatibility, fiber morphology control, material compatibility, limited fiber orientation, and mechanical strength.

## 4. Nanoparticle-Embedded ODFs

The advantages of embedding either polymeric or lipid nanoparticles into ODFs are multifaceted, including enhanced drug solubility and permeability, controlled release, targeted drug delivery, taste masking, and improved stability during storage and transportation [130,131,132]. Embedding nanoparticles into ODFs usually requires specialized formulation methods like solvent casting, HME, and spray drying. These techniques offer the necessary control and compatibility for successfully embedding nanoparticles in ODFs while addressing the challenges associated with their integration into the film formulation. A schematic diagram depicting the typical nanoparticle types and methods employed for incorporating nanoparticles into ODFs is presented in Figure 6. These techniques are employed to achieve the even dispersion of nanoparticles while preserving the film’s mechanical characteristics and overall integrity. Despite the promising potential of nanoparticle-loaded ODFs, there are challenges to address, including concerns about nanoparticle toxicity, regulatory compliance, and quality control. It is essential to thoroughly evaluate these aspects to ensure the safety and efficacy of such formulations. Several studies have suggested that formulating APIs in lipid nanoparticles can enhance their oral bioavailability [133,134]. Additionally, lipid formulations have been shown to mitigate the impact of food on API absorption [135]. The main approach to improving the bioavailability of APIs in lipid dispersions revolves primarily around creating surface-active monoglycerides. These monoglycerides, in combination with bile salts, give rise to mixed micelles that can encapsulate drug molecules. These mixed micelles are believed to facilitate the direct absorption of the API in conjunction with the lipids.

In addition, the release of bile salts can potentially improve the solubility of orally administered inadequately aqueous soluble APIs, thus facilitating their immediate absorption. While there have been limited studies on integrating solid lipid nanoparticles into mucoadhesive films, there has been even less focus on incorporating lipid microparticles into ODFs [136]. In a recent study [137], researchers explored the potential of utilizing HPMC film as a promising vehicle for solid lipid particles. Their findings demonstrated that the lipid content in the film matrix could reach as high as 54 wt.%, depending on the specific type of triglyceride nanoparticles integrated. To create lipid dispersions for inclusion in ODFs, a high-pressure homogenization process was employed, with formulations composed of 10% lipid, 5% stabilizer, and 0.5% sodium dodecyl sulfate (SDS). Some formulations solely relied on the surfactant SDS for stabilization, with a composition of 10% lipid and 0.5% SDS. During the evaluation of the stabilizers, various particle formulations were tested for three different triglycerides (tristearin, tripalmitin, and trimyristin). The most promising outcomes were observed with combinations of a polymer and the surfactant SDS, as well as formulations containing only SDS. Two specific particle formulations, HPMC/SDS and SDS, stood out as highly promising due to their ability to achieve substantial lipid contents and produce high-quality films. For tristearin, lipid contents of up to 0.50 were achievable in the ODFs with HPMC/SDS stabilization, and they were up to 0.54 for the SDS-based formulation. It is worth noting that the maximum lipid load was lower for the films loaded with tripalmitin (0.40) and trimyristin (0.29) compared to tristearin. A noteworthy aspect of this study was the successful redispersal of lipid nanoparticles from the film matrix without compromising their nanoparticulate properties. Additionally, the films exhibited commendable mechanical properties and disintegration times. Moreover, this study demonstrated the feasibility of substantially stabilizing lipid particles in the metastable α-polymorphic form for a duration of at least 6 months when utilizing HPMC/SDS to stabilize the triasterin suspension [137]. This breakthrough discovery opens up new possibilities for leveraging HPMC films to enhance the delivery of lipid-based drug formulations.

One innovative technique involves integrating a self-micro emulsifying system into the film, as previously explored [138]. However, this method demands a substantial quantity of oil, posing a challenge to the long-term integrity and stability of the film. To address this concern, another strategy was employed, i.e., reducing the drug to the nano-scale before its incorporation into ODFs. Research has shown a remarkable 11-fold increase in the in vitro drug release from polymeric films embedded with nanoparticles in comparison to the pure drug [139]. Other researchers have reported identical improvements in ex vivo permeation, emphasizing the substantial promise of using nanoparticle-loaded ODFs for the delivery of poorly soluble compounds [140,141]. In a recent study, scientists employed the Box–Behnken method to design and fabricate ODFs loaded with ketoprofen nanoparticles [142]. The amorphous state of drug-embedded nanoparticles within the ODFs was verified through the absence of distinct crystalline patterns and the absence of endothermic peaks of ketoprofen in X-ray diffraction and modulated differential scanning calorimetry analyses, respectively. The optimized formulation demonstrated a nearly fourfold increase in permeability compared to pure drugs. Furthermore, the dissolution rates of the drug and the optimized drug-containing ODF in a pH 1.2 solution reached approximately 30% and 95% at the 60 min mark, respectively. Pharmaceutical nanosuspensions are necessary when dealing with problematic drug molecules that cannot form salts, have high molecular weights, require a large dose, possess high log *p*-values, and have high melting points, as these factors make it difficult to create suitable drug formulations using other methods. A variety of techniques are employed to create nanosuspensions with diverse particle sizes. These methods include top-down approaches like dry and wet milling, co-grinding, and high-pressure homogenization, as well as bottom-up techniques such as antisolvent precipitation, liquid emulsion, and sonoprecipitation methods [143]. Nanosuspension-based oral thin films offer a promising approach to drug delivery, addressing the challenges associated with solubility, bioavailability, and patient compliance for a variety of pharmaceutical compounds. Fast-dissolving oral films of buspirone were created by employing the solvent evaporation method to transform a nanosuspension with the film-forming agents HPMC E5 and PVA. Buspirone oral films exhibited remarkable physical and mechanical characteristics and displayed good stability, and the in vitro assessments revealed an initial rapid drug release followed by a sustained release [21]. The inclusion of nanoparticles in oral films was anticipated to improve the dissolution and permeability properties of various poorly water-soluble drugs. A fast-dissolving oral film containing the poorly aqueous soluble and low bioavailable drug, lercanidipine, was fabricated as nanoparticles through the antisolvent evaporation technique. These formulations demonstrated a substantial enhancement in the in vitro dissolution rate and ex vivo permeation [21].

Pharmacokinetic investigations involving lutein nanocrystals in fast-dissolving oral films in rats demonstrated a significant reduction in the Tmax and a substantial increase in the Cmax compared to the oral solution. Additionally, the AUC_0–24_ h of nanocrystal fast-dissolving oral films was approximately two times larger than that of the oral solution, confirming a substantial enhancement in both the rate and extent of bioavailability [144].

A research study explores the application of the wet-milling technique to produce the nanosized hydrophobic drug loratadine, resulting in enhanced solubility and quicker dissolution [145]. These nanosized materials (<400 nm) were then employed in the formulation of ODFs that disintegrate rapidly, typically in less than 60 s. Although the drug’s crystalline structure remains unaltered, there is a significant improvement in its bioavailability. When administered to rats, the nanocrystal ODF leads to a 5.69-fold increase in the AUC_0–24_ h compared to the original drug, with a faster onset of action, typically within 30 min. The simplification of the formulation process and the enhancements in drug solubility and bioavailability have positioned the ODF as a promising drug delivery method for loratadine. In another interesting investigation, poorly aqueous soluble drugs such as fenofibrate and naproxen were formulated as nanoparticles using various strategies and were incorporated into ODFs [146]. The research findings revealed that the dose of API that can be administered in a single ODF significantly relies on the chosen formulation approach and the physicochemical characteristics of the API. The amorphous solid dispersion-ODFs and films incorporating API-loaded lipid nanoemulsions emerged as the most favorable film formulations. These formulations resulted in a decrease in the API dissolution time when compared to ODFs containing non-formulated API microparticles. Despite slightly compromised mechanical film properties compared to API-free film formulations, these ODFs achieved rapid disintegration times. For naproxen in the solid dispersion-ODFs, an API loading of up to 8 wt.% was attained without encountering recrystallization. In contrast, during the formulation of FENO in the films, the API content was limited to 2 wt.%, indicating unique intermolecular interactions between the APIs and the film-forming matrix.

## 5. In Vitro and In Vivo Evaluation Methods

The assessment of ODFs involves a range of essential tests to ensure their quality and performance. These tests encompass measurements of thickness, weight consistency, film durability, flexibility, water absorption, swelling, surface characteristics, and moisture content. Evaluating the taste perceptibility of an ODF is crucial, as it significantly influences patient acceptance and adherence to the medication. Additionally, mechanical properties such as tensile strength, puncture strength, elongation at break, elastic modulus, porosity, and folding endurance are evaluated, typically following the ASTM D882-01 standard [147]. The microstructural, compositional, and thermal characteristics of the films were examined by employing a range of analytical techniques, including scanning electron microscopy, X-ray diffraction, differential scanning calorimetry, and Fourier-transform infrared spectroscopy. Numerous investigations have underscored the impact of the disintegration evaluation method on the final results of gelatin-based ODFs. In the case of the slide frame method, where a solution drop is placed on the film’s surface, and the time for the solution to dissolve the film and form a hole is measured, it has demonstrated favorable disintegration times of around 30 s in multiple research studies [54]. In contrast, when using the Petri dish method, the sample is immersed in a solution, and the time taken for the film to fully disintegrate is documented. Research conducted with this approach has documented disintegration times exceeding 5 min [55]. Table 4 offers a comprehensive compilation of frequently utilized in vitro techniques for the evaluation of ODFs. These techniques are invaluable in pharmaceutical formulation development for researchers to assess the quality, performance, and characteristics of ODFs in a controlled laboratory setting. The schematic representation of the texture analyzer, Franz diffusion cell, dissolution apparatus type II, and dissolution apparatus type V, utilized for the in vitro evaluation of ODFs, is presented in the Figure 7, Figure 8 and Figure 9.

**Table 4 pharmaceutics-15-02753-t004:** Frequently used in vitro techniques in the evaluation of ODFs.

Technique	Principle	Evaluation Parameters	Ranges Units	References
Mechanical characteristics				
Tensile strength	Measure of a film’s ability to resist being pulled apart or stretched under tension without breaking or undergoing permanent deformation.	Tensile strength = Maximum load (N)/Original cross-sectional area (cm^2^) of the film	1–33 Megapascals (MPa)	[64]
Young’s modulus	Also known as the elastic modulus or modulus of elasticity, it measures the stiffness, toughness, or rigidity of a material.	Young modulus = Stress (σ)/Strain (ε)	50–1800 MPa	[138,148]
Percentage at break	Also known as elongation at break or strain at break; it is a measure of the amount of deformation a material undergoes before it fractures or breaks.	Strain at break = [(Final length − Original length)/Original length] × 100	0.3–38%	[148,149]
Strain energy	Elastic energy or deformation energy is the energy stored within a material when it undergoes deformation or strain.	Strain energy = (1/2) × Stress (σ) × Strain (ε) × Volume (V)	Joules (J) or Newton-meters (N·m)	[15]
Fracture energy or toughness	Measure of the amount of energy needed to cause complete failure or fracture of the material.	Energy to break = ∫ Stress (Applied stress) × d Strain (incremental change in strain)	J or N·m	[9]
Puncture strength	Also referred to as puncture resistance and is a measure of a material’s ability to withstand the penetration or puncturing force applied to it without rupturing or tearing.	Material’s resistance to the formation of holes or punctures or toughness	0.2–13 mJ	[150]
Indentation test	Measures the hardness or material response to localized deformation; it involves applying a controlled force or load to the surface of a material using an indenter, typically a hardened ball or a diamond tip, and measures the resulting depth or indentation formed.	Hardness and Young’s modulus	1 mPa and ~100 mPa	[151]
Folding endurance	Refers to the number of times the sample strip of the film can be folded front and back at a 180° angle at the same plane without breaking.	Flexibility and durability	~300 count	[152]
Film properties				
Water absorption capacity	The swelling capacity of an oral film evaluates its bioadhesion behavior, drug release, and physical integrity by measuring its ability to absorb water and expand in size or volume.	The percentage hydration of an oral film is determined using the equation [(X2 − X1) × 100/X1], wherein X1 represents the initial weight of the film, and X2 represents the weight of the film after hydration in simulated salivary fluid for a specified duration.	5–25%	[153]
Thickness	The thickness of the film is inversely proportional to its disintegration time. The thinner films are generally more comfortable for patients and can affect the rate of drug release and absorption. Maintaining a consistent thickness is crucial for ensuring content uniformity across different dosage units.	Thickness measurement is carried out at the center and four corners of the film using a digital micrometer or vernier calipers.	50–500 µm	[154]
Surface area	The surface area of the film can impact the rate of disintegration and dissolution of the API, which may contribute to absorption and bioavailability.	The surface area depends on its shape calculated using the appropriate geometric formula. For a square film, the total surface area is determined by squaring the length of any of its sides, and to determine the overall surface area of a rectangular film, simply multiply its length by its width.	2–5 cm^2^	[154]
Weight variation	Weight variation estimates the degree of variation or deviation in the weight of individual film units cut from the center and four corners within a batch or sample.	Weight variation is determined by calculating the difference between the weight of each film unit and the average weight of the film batch and then dividing it by the mean weight of the film.	<50 mg	[154]
Content uniformity	It ensures that the API is distributed uniformly throughout the dosage unit, minimizing variability between individual units.	Reliable and validated analytical method for the quantification of API. The individual film unit conforms to USP <905> uniformity of dosage units.	-	[154]
Surface characterization				
Surface morphology	Cutting a small section of the film, affixing to stubs utilizing adhesive tape, sputter-coating with a conducting material in an inert gas, and scanning it in a scanning electron microscope.	Surface features, roughness, presence of pores, cracks, surface irregularities, thickness, particle size, and other surface-related properties.	-	[155]
Surface pH	Subjected to a brief (<2 h) contact with distilled water at a temperature of 37 ± 0.5 °C, facilitating its swelling. It provides information about the acidity or alkalinity of the film’s surface	pH at the area of application, potential irritancy, or compatibility of the film with the oral tissues.	6.0–7.5	[156]
Film analysis				
Crystallinity and amorphous nature	Test material is placed in the ample holder of the X-ray diffractometer and scanned using specific voltage and current. Diffraction patterns were obtained at various 2θ values at a uniform scan speed with a particular step width.	Provide valuable information about the composition, crystallinity, amorphous nature, and physical properties of the sample.	%	[157]
Thermal scanning	The sample was sealed in the aluminum pan and heated at a uniform rate within the specified temperature range under an inert nitrogen atmosphere.	Examine the presence of phase transformation, recrystallization, thermal properties, and material characterization within the film.	°C	[158]
Infrared spectroscopy	Test sample and potassium bromide compressed at a certain pressure and scanned, or ground solid sample into a fine powder or placed into a liquid sample on an appropriate IR-transparent substrate, such as a salt plate or a diamond crystal.	Compatibility between drug and various inactive pharmaceutical ingredients in the formulation.	cm^−1^	[159]
Release properties				
Disintegration test	Tested in 10 mL of simulated salivary fluid (pH 6.4) placed in a beaker maintained at 37 ± 0.5 °C and stirred at 10 rpm.	Assess the film’s ability to disintegrate/dissolve when in contact with simulated salivary fluid or other suitable fluid.	30–60 s	[160]
In vitro drug release	Synthetic membrane using Franz diffusion cell, or utilizing USPXXIV Type 2 apparatus, USP Type V paddle-over-disk apparatus.	Drug release from the developed film using simulated salivary fluid (pH 6.2) or other suitable media. Acceptance criteria are similar to immediate-release conventional dosage forms.	%	[161,162]
Ex vivo permeation	Investigation carried out on freshly isolated buccal mucosa obtained from rabbit, pig, or chicken utilizing Franz diffusion cell, continuous flow through diffusion cell, Ussing chamber, or cell culture model using human oral epithelial cell line (TR-146).	Understand the transportation mechanism of a drug through oral epithelial tissues using parameters such as flux and permeability coefficient.	Flux (J) = μg/cm^2^/h permeability coefficient (P) = cm/h	[162]
Biocompatibility				
Cell cytotoxicity	Assess the potential toxicity of a substance or material on living cells. In vitro model of buccal mucosa, TR-146 cells cultured in Dulbecco’s modified eagle’s medium in optimum conditions at 37 °C in an atmosphere of 5% CO_2_ in plastic cell culture flasks with minimum passages of 20 and 28. Other techniques include MTT assay, lactate dehydrogenase release assay, live/dead staining, and cell proliferation assays.	Determine the impact of the ODF on cell viability, cell membrane integrity, and overall cellular function.	Number of viable cells	[163]
Organoleptic properties				
Taste perception	Tribology tests using a Bio-Tribometer aim to assess friction in a simulated tongue–palate interaction, taking into account the mechanical factors in the oral cavity and utilizing artificial (polydimethylsiloxane) models of biological tissues such as the tongue, palate, and saliva as test samples. A toolbox of techniques comprising Petri dish and oral cavity model methods for testing disintegration time and bio-tribology tests for the disintegration and oral perception of ODFs.	The test mimics the motion of the tongue and palate by employing a blend of a wide contact area and a relatively brief stroke length. Combined techniques to enable formulators to create, evaluate, and reformulate ODF.	Coefficient of friction (N)	[164]
Sensory panel studies	Perception of taste is a complex and intricate phenomenon characterized by individual variations in sensitivity and response, influenced by various physiological and environmental factors.	Each film strip is placed on the tongue and examined for a duration of 10 s using the swirl and spit technique. Standardized neutralization is conducted between successive samples. Bitterness intensity is promptly evaluated on an 11-point scale, ranging from 0 (not bitter) to 11 (bitter)	Bitterness score	[165]
Electronic tongue	It serves as a taste sensor equipped with a set of multichannel detectors designed to replicate the conditions of taste buds found on human tongues.	It assesses taste by gauging the detectors’ response to diverse compounds, delivering a swift and impartial taste evaluation. The evaluation parameters typically include factors such as taste intensity, sweetness, bitterness, sourness, and sometimes additional characteristics like saltiness or umami.	Bitterness score	[166]

The evaluation of ODFs extends beyond in vitro tests, with the use of animal models for predicting the in vivo absorption. Allometric equations facilitate the dose conversion between humans and animals [169]. In vivo studies often rely on pharmacokinetic parameters like AUC, Tmax, and Cmax to estimate the drug absorption of nanoparticles, which can be tracked and assessed in tissues using fluorescent labeling. Nevertheless, it is crucial to acknowledge that the plasma drug concentration cannot differentiate between absorption from the pregastric and intestinal regions [4]. Ex vivo mucosal permeation models, which closely mimic the histological features of the buccal membrane, offer more accurate predictions of the in vivo drug absorption compared to an evaluation employing artificial membranes [170]. These models provide valuable insights into the intricate processes governing nanoparticle transport, including their interactions with mucus barriers. 

Patients are more likely to adhere to their treatment regimens when the taste of the ODF is palatable, making it a key factor in the overall therapeutic experience. Therefore, conducting sensory assessments to understand the taste characteristics and how they affect the patient’s willingness to use the product is essential in pharmaceutical and healthcare research. There are currently limited testing methods capable of accurately evaluating the sensory aspects that affect the acceptability of ODFs. In the reported study, four ODFs were fabricated using PVA and NaCMC as the primary components through the 3DP method [171]. The in vitro disintegration times were tested using the Petri dish, oral cavity model, and bio-tribology methods. The data showed that an increase in the molecular weight of the polymers had an exponential impact on the time it took for the films to disintegrate in the Petri dish and oral cavity. Moreover, polymeric films with higher molecular sizes exhibited a stronger adhesion to the upper palate in the oral cavity method. Bio-tribology analysis revealed that films with a higher molecular weight disintegrated more rapidly and had a lower coefficient of friction, suggesting potentially favorable oral perception but also a certain stickiness due to increased viscosity. These evaluation techniques have the potential to assist formulators in designing, testing, and reformulating ODFs that can achieve rapid disintegration and offer improved sensory attributes when consumed, thereby enhancing overall treatment efficacy and acceptability. In summary, the assessment of oral thin films involves a comprehensive battery of tests to ensure their physical and mechanical properties, as well as their suitability for drug delivery. While the in vitro evaluations are crucial, ex vivo and in vivo models using animal and mucosal tissue provide valuable insights into real-world drug absorption and biodistribution.

## 6. Clinical Translation and Future Prospects

ODF is applicable for local delivery in mouth conditions, such as a local anesthetic in dental procedures, bacterial or fungal infections, sore throats, oral ulcers, pharyngitis, or tonsillitis [7]. ODF offers numerous benefits, making it particularly advantageous in emergency scenarios and for patients with schizophrenia and dysphagia. ODFs prepared with small molecules were successfully developed for local as well as systemic effects [73,172]. Table 5 offers a concise overview of the clinical trials, both ongoing and completed, pertaining to various drug-loaded ODFs designed for oral delivery. Back in 2010, the FDA authorized the first prescription ODF, known by the trade name Zuplenz, and contains ondansetron as its API. Since then, many prescription and non-prescription ODFs have been approved and have gained popularity worldwide. Nonetheless, numerous formulation hurdles still exist, including the challenging issues of unpalatable-tasting drugs and high dosages. The feasibility of ODFs for the delivery of nanoparticles has been extensively explored [173,174].

Many companies have secured patents for their ODFs, and Table 6 lists some examples of these patented ODFs, along with a summary of their innovation. An in-depth examination of patent technologies employed in the production of ODFs has been recently reviewed [175]. Table 7 highlights an overview of selected commercial examples of ODFs and their respective therapeutic categories. This information offers valuable insights into the diversity of ODFs in the market and the therapeutic areas they are primarily designed to address.

ODFs are typically categorized into three classes based on different criteria, such as dissolution properties, layering characteristics, and the nature of the API. Dissolution-based ODFs can be categorized as rapidly dissolving, with dissolution occurring in less than 30 s; moderately dissolving, taking between 1 to 30 min; or slowly dissolving, requiring more than 30 min to fully dissolve. Monolayer ODFs generally include an API, a film former, a plasticizer, and inactive ingredients. On the other hand, bilayer or oral films have an API layer along with an additional taste-masking or mucoadhesive layer, acting as a separator between the two. Multilayer films employ an API layer positioned between two layers, making this approach beneficial for effectively combining incompatible drugs [176]. The widely used approach for creating ODFs involves the solvent casting method, wherein active substances are dissolved or dispersed in biocompatible polymeric films, followed by evaporation. However, there is a growing interest in 3DP methods, including HME, FDM, and inkjet techniques. ODFs present certain drawbacks in drug delivery, including limitations in accommodating a substantial API load, challenges in taste masking, and potential variations in the uniformity of the content. Additionally, there may be limitations related to stability, shelf life, and susceptibility to environmental factors. These considerations underscore the importance of addressing multiple facets in ODF formulation to enhance overall efficacy and patient acceptance. The primary constraint linked to film dosage forms lies in the challenge of attaining a substantial payload within the restricted surface area. One possible approach to addressing this problem is to encapsulate a high payload of active substances in nanoparticles and then embed these nanoparticles in mucoadhesive polymeric materials. Presently, ongoing research is focused on nanoparticle-enabled films, investigating the diverse functionalization approaches aimed at improving diffusion through the oral mucosa and facilitating the systemic targeting of actives. The main issue resulting in the non-compliance of patients is the taste of drugs. However, the unpleasant taste of the drug in the film dosage form can be partially masked by using certain agents, such as a complexing agent (e.g., β-CD), a cooling agent (e.g., menthol), and sweetening agents like mannitol and aspartame sodium.

Numerous pharmaceutical companies are transitioning their product lines from ODTs to ODFs. Several APIs are being used for ODFs, targeting conditions like Parkinson’s disease, depression, schizophrenia, and Alzheimer’s disease. The identification of novel functional excipients with the capability to enhance penetration and the investigation of new approaches for oromucosal permeation, such as the ion-pair strategy, represent promising paths for advancing ODFs. The future of research and development in the field of ODFs may focus on 3DP techniques, which offer the potential to incorporate high drug loading, explore combinations of multiple drugs, and introduce compartmentalization for isolating incompatible drug components.

## 7. Conclusions

ODFs represent a versatile and patient-friendly dosage form that has gained significant attention in the pharmaceutical industry. ODFs offer numerous advantages, such as rapid drug delivery, improved patient compliance, and suitability for various drug types, including poorly soluble compounds and macromolecules. Recent advancements in the formulation of ODFs, including the incorporation of nanoparticles, have opened new avenues for enhancing drug delivery efficiency and expanding the range of drugs amenable to this administration route. However, further research and development are still needed to address the challenges related to taste masking, stability, and the incorporation of complex active ingredients. ODFs continue to be an exciting area of pharmaceutical innovation, with the potential to revolutionize drug delivery for a diverse spectrum of therapeutic uses. As the discipline progresses, ongoing efforts in formulation science, regulatory guidance, and clinical evaluation will be critical to fully realizing the possibilities of ODFs in improving patient outcomes and healthcare services.

## Figures and Tables

**Figure 1 pharmaceutics-15-02753-f001:**
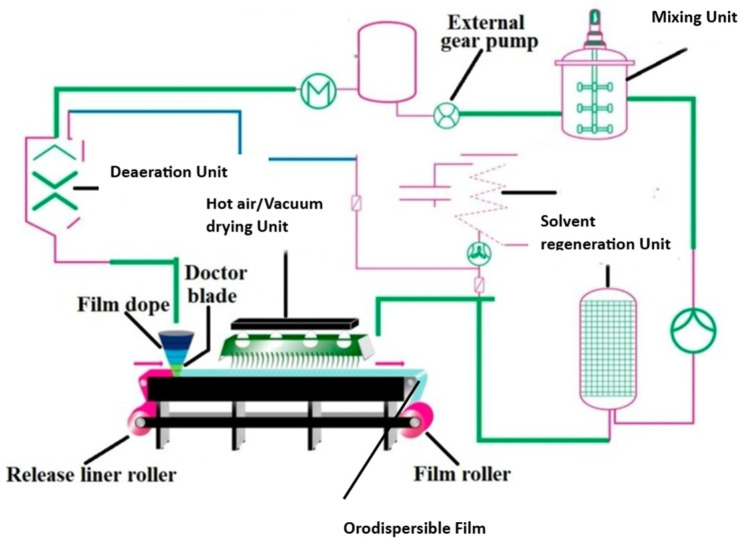
Schematic diagram displaying the typical solvent casting processes involved in orodispersible film manufacture (adapted from [4], published by MDPI, 2021).

**Figure 2 pharmaceutics-15-02753-f002:**
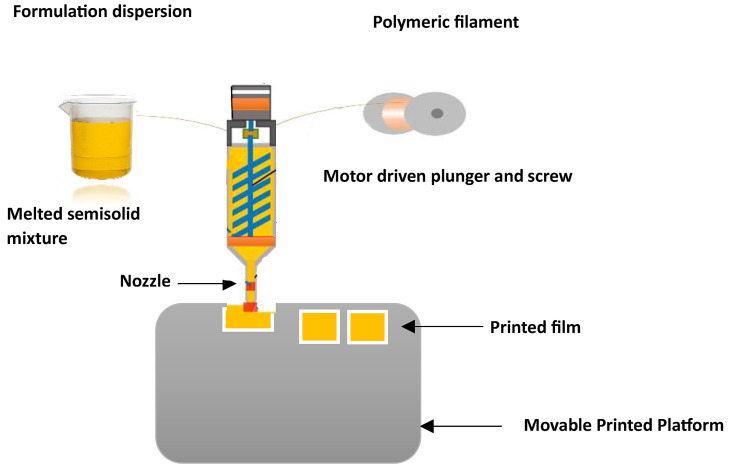
Illustration depicting ODF 3D printing based on the semisolid extrusion process.

**Figure 3 pharmaceutics-15-02753-f003:**
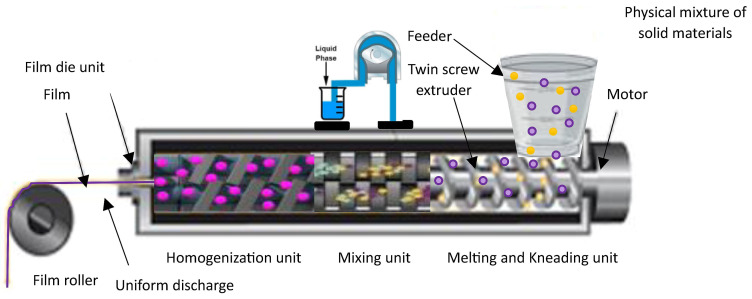
Manufacture of ODF utilizing hot-melt extrusion technique.

**Figure 4 pharmaceutics-15-02753-f004:**
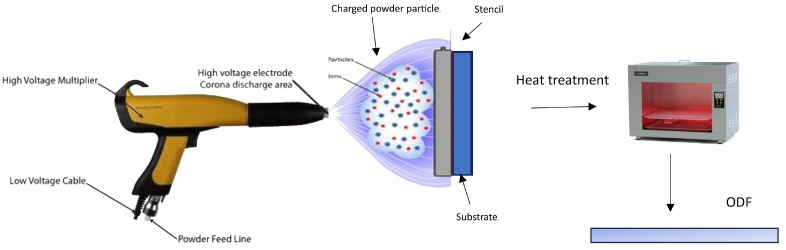
ODF fabrication based on electrospray powder deposition process.

**Figure 5 pharmaceutics-15-02753-f005:**
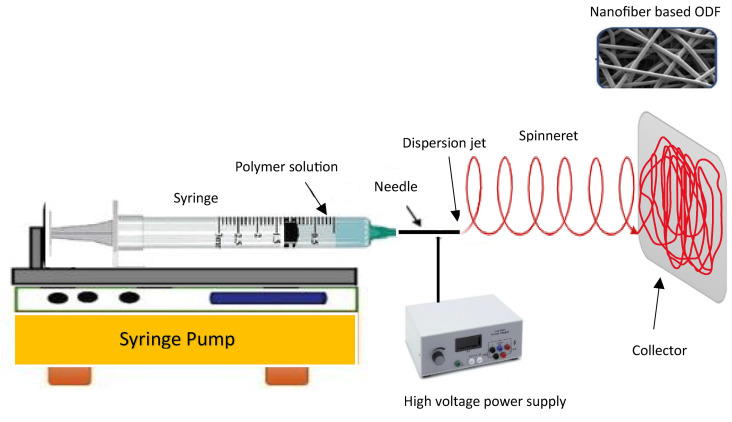
Steps involved in the electrospinning process for preparing ODFs.

**Figure 6 pharmaceutics-15-02753-f006:**
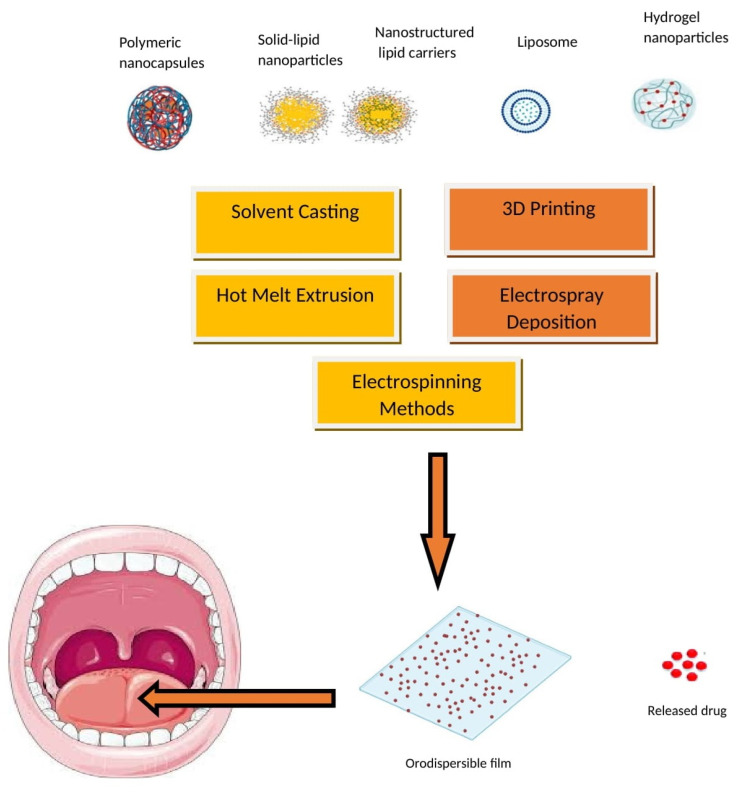
Nanoparticle types and incorporation methods in ODFs.

**Figure 7 pharmaceutics-15-02753-f007:**
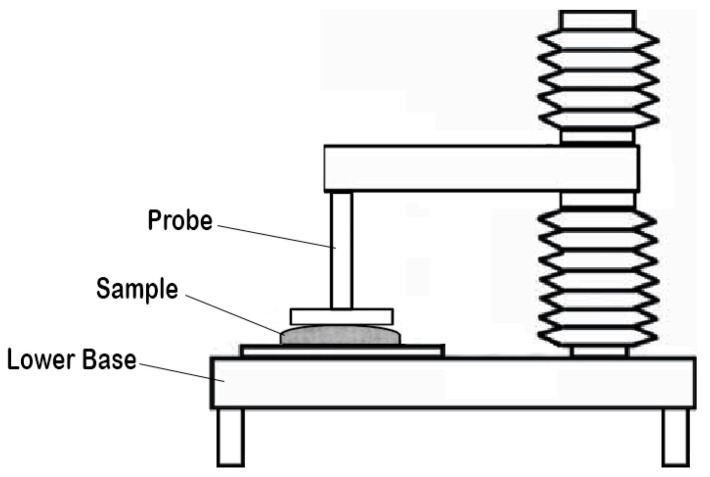
Schematic diagram of the texture analyzer (adapted from [167], published by MDPI, 2020).

**Figure 8 pharmaceutics-15-02753-f008:**
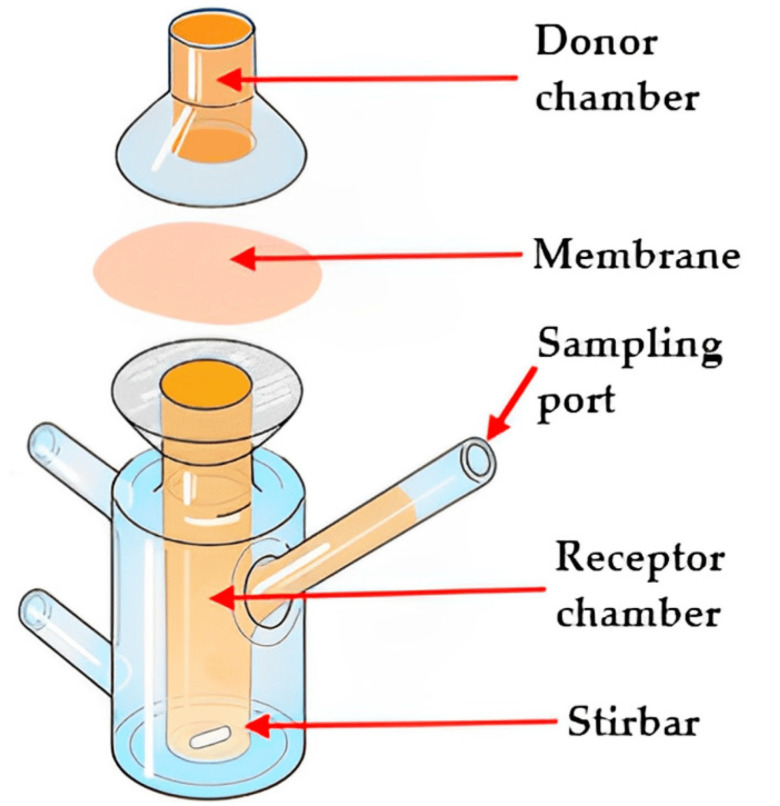
Illustration of the Franz diffusion cell (adapted from [168], published by MDPI, 2023).

**Figure 9 pharmaceutics-15-02753-f009:**
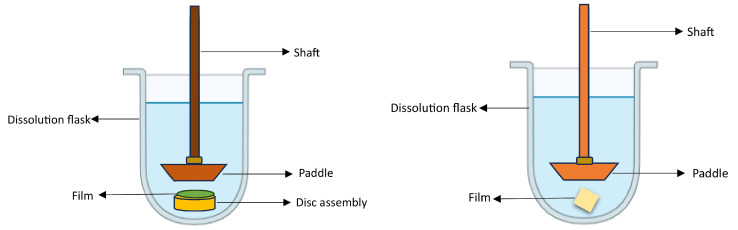
Schematic illustration of the USP dissolution apparatus type II (**left**) and dissolution apparatus type V (paddle-over-disk (**right**)).

**Table 1 pharmaceutics-15-02753-t001:** Comparison of advantages and drawbacks in orodispersible film drug delivery systems.

Aspects	Advantages	Drawbacks
Ease of administration	Convenient, easy, and safe for administration	May not be suitable for certain patient populations experiencing swallowing difficulties
Patient compliance	Enhanced patient compliance, particularly for geriatric, pediatric, disabled, and dysphagic populations	Formulation might possess a bitter taste and may include allergens or excipients, potentially affecting patient acceptability
Rapid onset of action	Quick disintegration	Limited capacity for delivering modified-release formulations
Dosage accuracy	Precise dosing	Restricted ability to accommodate high-dose medications
Bioavailability	Improved bioavailability due to potential oromucosal absorption	Stability concerns, especially for certain drug formulations
Dosing	Feasibility of personalized dosing	Storage and handling requirements may be more stringent
Market acceptance	Emerging popularity in the market	Expensive compared to conventional dosage forms

**Table 2 pharmaceutics-15-02753-t002:** Generic critical quality attributes for an orodispersible film (ODF).

Critical Quality Attributes		Target
Physical attributes	Size	Length, width, and thickness allow the film to be convenient for placement on the surface of the tongue. The size of the film is 1 cm × 1 cm, and the thickness is 100 µm
	Mechanical characteristics	High tensile strength, high elongation at break, and low Young’s modulus
Identification		Positive for drug
Assay		100% *w*/*w* of label claim
Content uniformity		Conforms to USP <905> uniformity of dosage units
Disintegration time		Not more than 60 s
Dissolution		Acceptance criteria similar to the conventional immediate-release solid dosage forms

**Table 3 pharmaceutics-15-02753-t003:** Different types of film formers, plasticizers, actives, preparation methods, and key highlights of ODFs.

Polymer/Plasticizer	Drug	Method	Highlights	Reference
Carboxymethyl cellulose and gelatin type A/D-sorbitol	Caffeine	Solvent casting	Gelatin type A proved to be more efficient in regulating caffeine release, with a peak value of 97.4% ± 0.95 occurring after 20 min. Conversely, carboxymethyl cellulose is more suitable for immediate release, as it resulted in a maximum caffeine release of 81.1% ± 2.14 within 4 min. The ex vivo permeability assessment was consistent with the USP dissolution assay, where 42.0% ± 7.79 of caffeine was released from carboxymethyl cellulose and 15.3% ± 4.0 from Gelatin type A oral films.	[15]
Carboxymethyl cellulose/sorbitol	Ora-pro-nobis extract	Tape casting technique	Fast disintegration (<50 s) combined with the release of iron, occurring within a maximum of 26 min in CMC-based ODF microparticles and 50 min with CMC-based ODF containing ora-pro-nobis (OPN) extract. Release kinetics indicated the Fickian diffusion mechanism Developed systems have the potential capacity to serve as innovative means for delivering oral iron intake, beneficial for iron deficiency.	[16]
Hydroxypropyl cellulose SL	Ibuprofen/Calcium carbonate	Solvent casting method	Disintegration time decreased as the film thickness decreased. The incorporation of ibuprofen into the film extended the disintegration time, whereas embedding calcium carbonate particles in the film reduced it. Drug loading led to a reduction in tensile strength as well as elastic modulus.	[17]
HPMC 603/PEG 400	Ropinirole	Solvent casting	Ropinirole reached systemic circulation in less than 15 min. Bioavailability was tremendously enhanced by higher Cmax from sublingual (188.9 ± 25.1 ng/mL) and buccal films (166.7 ± 55.2 ng/mL) compared to the oral tablet (29.9 ± 16.2 ng/mL). Physical stability was demonstrated for at least 28 days.	[18]
HPMC E5/glycerin	Atenolol	Solvent casting	HPMC-based ODF demonstrated superior attributes in terms of uniformity, drug content, and dissolution properties. Molecular docking results, confirmed through FTIR spectral studies, indicated the presence of hydrogen bonding between atenolol and the film formers. pH measurement indicated neutrality, suggesting that the films would not sensitize oral mucosa.	[19]
HPMC E5/glycerin	Flupentixol dihydrochloride	Solvent casting method	ODF containing 2% HPMC demonstrated remarkable consistency and stability of flupentixol aside from rapid disintegration. ODF of flupentixol exhibited a rapid absorption rate, resulting in a relative bioavailability of 151.06% in comparison to the marketed product.	[20]
HPMC E5 and PEG 400/propylene glycol	Quetiapine fumarate	Solvent casting method	The mean thickness of the film was between 0.159–0.172 mm. The dry film thickness exhibited an inverse correlation with both the plasticizer quantity and drying temperature. Estimated disintegration times were in the range of 128.0–136.3 s. ODFs with higher propylene glycol and those dried at elevated temperatures demonstrated a notably faster dissolution rate when compared to other films.	[21]
HPMC E15/glycerin	Phenytoin	Syringe extrusion 3D printing	Optimized phenytoin-loaded ODFs displayed good physical appearance, acceptable mechanical strength, rapid disintegration (<5 s), and immediate drug release (80% < 10 min). Additionally, an enhancement in the solubility and dissolution rates was observed.	[22]
HPMC E15 and PVA (Polyvinyl alcohol)	Buspirone hydrochloride	Solvent casting method	The nanoprecipitation technique was utilized to prepare poly(lactic-co-glycolic acid) nanoparticles and embed them into the polymeric dispersion-containing plasticizer. Particle size, zeta potential, and entrapment efficiency of the nanoparticles measured were 189.23 ± 0.95 nm, −21.56 ± 0.56 mV, and 68.28 ± 3.69%, respectively. XRD studies showed the existence of the drug in an amorphous state. The in vitro drug release demonstrated prolonged delivery for 48 h and followed the Korsmeyer–Peppas model.	[21]
HPMC K100/glycerin	Topiramate	Solvent casting	The D-optimal statistical experimental design showed that independent variables related to polymer and plasticizer had a predominant impact on dependent variables like disintegration time, assay, and film thickness. The fabricated film released 98% of topiramate in just 10 min while preserving the drug’s physicochemical stability.	[23]
HPMC LVE3/glycerin	Venlafaxine HCl	Solvent casting	The film formula with moderate levels of HPMC and glycerol, along with the highest concentration of sodium starch glycolate, resulted in the highest observed swelling index at 3.64 ± 0.59. Optimized ODF composition, based on the statistical approach, indicated 2% HPMC, 5% sodium starch glycolate, and 1% glycerol.	[24]
Maltodextrins/sorbitol	Benzydamine hydrochloride	Semisolid extrusion 3D printing method	Modifying the formulation’s viscosity enabled direct printing into thin layers, followed by in-process drying. The dose could be controlled by adjusting the thickness or overall volume of the digital model. The modified printing method holds potential for personalized film dosage production and compartmentalization of drugs aside from the potential for taste masking or release control ability.	[25]
Maltodextrins/glycerin	Quercetin	Modified solvent casting method	Quercetin nanosuspensions were prepared by high-pressure homogenization. The average particle size was 753 nm, and the polydispersity index was 0.31. The dissolution profile of the drug, loaded in fast-dissolving film, was significantly faster than that of the free freeze-dried nanocrystals.	[26]
Modified starch/glycerin	Chlorpheniramine maleate	Hot-melt extrusion technology	Glycerol lowers the melt viscosity and enhances the free volume of starch chains, thereby facilitating the extrusion process. The absence of peaks in melt-extruded formulation confirms the presence of the drug in an amorphous state. ODF formulations created via hot-melt extrusion technology exhibited rapid disintegration times (6 to 11 s), achieving over 95% dissolution within 5 min. The human panel study and animal model confirmed a reduction in bitterness in the films compared to the pure drug and physical mixture.	[27]
Polyvinyl alcohol 4 -88/PEG 400	Meclizine hydrochloride	Solvent casting	The thickness and tensile strength of the meclizine ODF was 0.116 ± 0.004 mm and 17.37 ± 1.54 N mm^−2^, respectively. Drug dissolution exceeded 80% in less than 5 min, both in distilled water and 0.1 M hydrochloric acid. ATR-FTIR analysis revealed that the drug molecules were incorporated into the polymer’s network structure, leading to the prevention of drug recrystallization. The Cmax values for Zentrip^®^ (Tokyo, Japan) and meclizine OTF were 1.46 ± 0.44 μg/mL and 1.91 ± 0.51 μg/mL, while their respective AUC values were 10.38 ± 2.93 μg h/mL and 13.74 ± 3.23 μg h/mL.	[28]
Pullulan/glycerin	Ovalbumin, β-galactosidase and Lysozyme	Solvent casting method	Loading ovalbumin had no significant impact on the mechanical characteristics of freeze-dried ODFs, while introducing it into air-dried ODFs resulted in a notable decrease in tensile strength. The stability of lysozyme remained unaffected by changes in the trehalose/pullulan ratio, whereas higher ratios led to improved stability of β-galactosidase. Freeze-drying was more suitable for process stability, and air-drying was better for storage stability.	[29]

**Table 5 pharmaceutics-15-02753-t005:** Clinical trial status for ODF formulations developed for systemic delivery.

Clinical Trials	Indication	Phase	Enrolment	Identifier
Comparative bioavailability studies of Riluzole orodispersible film (32.0 mm × 22.0 mm) vs. marketed Rilutek^®^ Tablets (Strasbourg, France) containing an equivalent dose of 50 mg.	Amyotrophic lateral sclerosis	1	54	NCT04819438
Single-center, single-dose, open-label study to assess the effect of a single 50 mg dose of Riluzole oral soluble film.	Amyotrophic lateral sclerosis	II	9	NCT03679975
Prospective, interventional, multi-center, randomized, double-blind, fixed-dose, parallel-group clinical study to demonstrate the efficacy and safety of sildenafil oral film (50/75/100 mg) in comparison with placebo.	Erectile dysfunction	III	600	NCT05490680
Open, monocentric, comparative, crossover study to evaluate bioavailability after a single oral dose of iron ODF vs. SiderAL^®^ FORTE (Pisa, Italy) capsules in healthy women.	Iron supplement	I	9	NCT05660200
Assess the pharmacokinetics and efficacy of insulin-loaded orally dissolved films.	Glucose management	1	7	NCT01446120
Pharmacokinetic study in volunteer smokers with various doses (0/2/4 mg) of oral nicotine film, assessing for peak blood nicotine concentrations, safety, and subjective effects.	Oral nicotine replacement therapies	Interventional	24	NCT02239770
Open-label, balanced, two-treatment, two-period, randomized study comparing ondansetron orally dissolving film strip with Zofran ODT.	Chemotherapy-induced and radiation-induced nausea and vomiting, postoperative nausea and vomiting	1 and II	48	NCT01217190
Open-label study to assess the safety and tolerability in opioid-dependent subjects maintained on a stabilized dose of Suboxone tablets or films.	Maintenance treatment of opioid dependence	II	249	NCT01666119
Investigate the efficacy of oral thin film loaded with cholecalciferol in patients undergoing hematopoietic stem cell transplantation.	Vitamin D replacement therapy	Interventional	24	NCT04818957
A single-dose, randomized, three-period, crossover comparative bioavailability study of aripiprazole oral soluble film 10 mg (test) vs. Abilify^®^ (Tokyo, Japan) 10 mg tablet (reference) in healthy male volunteers.	Schizophrenia	I	36	NCT02501109

**Table 6 pharmaceutics-15-02753-t006:** A compilation of recently filed patents for ODFs and highlights of their innovation.

Application ID	Title	Composition of Invention	Date of Publication
US 20230136398 A1	Oral thin film with smooth fused film.	Comprising at least 40% *w*/*w* polyethylene oxide as a film-forming polymer with water-insoluble actives and utilizing solvent evaporation technique.	4 May 2023
US 20230133317 A1	Taste-masked and rapidly disintegrating ultra-thin iron orodispersible film and a process thereof.	Consists of microencapsulated iron, beta-cyclodextrin, flavoring agent, calcium carboxy methyl cellulose, pullulan, mannitol, sweetening agent, polyethylene glycol, lecithin, malic acid, ascorbic acid, and kiwi flavor.	4 May 2023
US 20220331337 A1	Orodispersible formulations.	Contains an active ingredient (0.5% *w*/*w*), an intragranular component (40–80% *w*/*w*) containing diluents, disintegrants, and binders, and an extragranular component containing diluents and disintegrants. ODF disintegrates within 60 s and has specific percentage ranges for each component.	20 October 2022
US 20170182105 A1	ODF.	Describes film-forming suspensions with a plant extract and. method for producing ODFs.	29 June 2017
US 20170165315 A1	ODF containing enalapril, designed for treating hypertension in pediatric patients.	Comprised of water-soluble polymer, preferably pullulan and modified starch such as Lycoat^®^ (Lestrem, France) (50–80% *w*/*w*), in addition to plasticizers, fillers, sweetening agents, and an acidic agent.	15 June 2017
US 20170143623 A1	Orodispersible films have quick dissolution times for therapeutic and food use.	Comprised of film-forming substances, such as maltodextrin (40–80% *w*/*w*), plasticizers (15–55% *w*/*w*), surfactant system (0.5–6% *w*/*w*), homopolymer or copolymer of vinyl acetate (1–20% *w*/*w*), and an active ingredient ranging between 0.05% and 30% *w*/*w*.	25 May 2017
US 11452698 B2	Dissolvable gel-forming film for delivery of active agents.	Presents a dissolvable, gel-forming film made up of a water-soluble cellulose ether and an active proteolytic enzyme.	27 September 2022

**Table 7 pharmaceutics-15-02753-t007:** Selected commercial examples of ODFs and their therapeutic category.

Product Name	Active/Key Ingredients (Dose)	Indications	Category	Manufacturer
Benadryl^®^ Allergy Quick Dissolve Strips	Diphenhydramine HCl (25 mg)	Allergic symptoms	OTC	McNeil-PPC, Inc., Fort Washington, PA, USA
Triaminic^®^ Children’s Thin Strips	Dextromethorphan (3.67 mg) and phenylephrine HCl (2.5 mg)	Cough suppressant and nasal decongestant	OTC	Novartis Consumer Health, Basel, Switzerland
Gas-X Thin Strips	Simethicone (62.5 mg)	Ant flatulent	OTC	Novartis Consumer Health, Basel, Switzerland
Pedia-Lax Quick Dissolve Strips	Sennosides (8.6 mg)	Occasional constipation	OTC	C.B. Fleet Company, Lynchburg, VA, USA
Risperidon HEXAL_ SF Schmelzfilm	Risperidone (5 mg)	Schizophrenia	Rx	Hexal AG, Holzkirchen, Germany
Sudafed_ PE Quick dissolve strips	Phenylephrine HCl (10 mg)	Nasal decongestant	OTC	McNeil-PPC, Inc., Fort Washington, PA, USA
Zuplenz^®^	Ondansetron (4/8 mg)	Antiemetics	Rx	Strativa Pharmaceuticals, El Segundo, CA, USA
NiQuitin	Nicotine (2.5 mg)	Nicotine replacement therapy	Rx	Boots UK Ltd., Nottingham, UK
Zolmitriptan ODF	Zolmitriptan (2.5 mg)	Migraine	Rx	APR Applied Pharma Research and Labtec, Balerna, Switzerland
Sildenafil Sandoz Orodispersible Film	Sildenafil (25/50/100 mg)	Erectile dysfunction	Rx	Sandoz, Basel, Switzerland
Sympazan oral film	Clobazam (5/10/20 mg)	Seizures associated with Lennox-Gastaut syndrome	Rx	Otter Pharmaceuticals, Lake Forest, IL, USA
IVYFILM	Ivy leaves dried extracts (16 mg)	Antitussives and expectorants	OTC	Forrester Pharma, Cape Town, South Africa
Chloraseptic Sore Throat Relief Strips	Benzocaine (3 mg)	Soothes throat and relieves pain	OTC	Prestige, Melville, NY, USA
Theraflu^®^ Thin Strips multi symptom	Diphenhydramine HCl (25 mg)	Allergic symptoms/Cough suppressant	OTC	Novartis Consumer Health, Basel, Switzerland
Theraflu^®^Thin Strips long acting cough	Dextromethorphan HBr (11 mg)	Cough suppressant	OTC	Novartis Consumer Health, Basel, Switzerland
Theraflu^®^ Daytime thin strips	Dextromethorphan (14.8 mg) and Phenylephrine HCl (10 mg)	Temporary relief for nasal and sinus congestion as may occur with a cold	OTC	Novartis Consumer Health, Basel, Switzerland
Theraflu^®^ Nighttime thin strips	Diphenhydramine HCl (25 mg) and phenylephrine HCl (10 mg)	Temporary relief for nasal and sinus congestion as may occur with a cold	OTC	Novartis Consumer Health, Basel, Switzerland
Triaminic^®^ thin strips cold with a stuffy nose	Phenylephrine HCl (2.5 mg)	Temporary relief for nasal and sinus congestion as may occur with a cold	OTC	Novartis Consumer Health, Basel, Switzerland
Triaminic^®^ Thin strips day time cold & cough	Dextromethorphan (3.67 mg)	Nasal decongestant	OTC	Novartis Consumer Health, Basel, Switzerland
Listerine PocketPak breath strips	Pullulan, xanthan menthol, methyl salicylate, eucalyptol, thymol, menthyl succinate	Mouth freshener	OTC	Johnson and Johnson Consumer Health Inc., New Brunswick, NJ, USA
Methylcobalamin orally disintegrating strips	Methylcobalamin (1500 µg)	Vitamin B12 supplement	OTC	Shilpa Medicare Ltd., Raichur, India
Tadalafil orally disintegrating strips	Tadalafil (10/20 mg)	Erectile dysfunction	Rx	D.K-Livkon Healthcare Pvt. Ltd., Mumbai, India
Levocetirizine orally disintegrating strips	Levocetirizine (5 mg)	Allergic symptoms	OTC	D.K-Livkon Healthcare Pvt. Ltd., Mumbai, India
Vitamin D3 orally disintegrating strips	Vitamin D3 (6000 IU)	Vitamin D3 supplement	OTC	D.K-Livkon Healthcare Pvt. Ltd., Mumbai, India

## Data Availability

The data presented in this study are contained within this article.

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
