# Peer review of "Orodispersible Films: Current Innovations and Emerging Trends"

_pharmaceutics, 2023, doi:10.3390/pharmaceutics15122753_

Round 1

Reviewer 1 Report

Comments and Suggestions for Authors

The authors review orodispersible films with specific focus on formulation, methods of preparation, evaluation, clinical studies and patent-based innovations and a table of commercialized products as examples.

Some parts of the subject can be found in other reviews. The strength of this review is to bring together various aspects, and to summarize the literature available in tables and figures, which are helpful for the reader.

To further improve the manuscript, here are my suggestions :

-       Lines 40-50 : to be consistent with what is said after for orodispersible films (line 65), please add local effect (eg : “enabling rapid absorption and delivery of the drug into the bloodstream” or rapid local effect)

-       In chapter Formulation components : a general table/figure, for example target product profile or generic composition with expected properties, could be helpful and complementary to the tables 1 and 2

-       Table 3 (general): the table make sense but it’s organization could be improved : for instance by making sub-sections (eg mechanical properties, surface characterization, release properties…).

-       Table 3 - Specific comments : for some items, the content of the column “Evaluation parameters” or “Range units” seems confusing – item "Thickness, area and weight variation" has for instance as evaluation parameter Content uniformity (which should be a separate item, but does not characterize the thickness, area and weight variation?). For the item "Taste perception", it is only about friction : what about the taste and its acceptability (eg bitterness scores etc)?

-       Figures for illustration of techniques are helpful – a comparison table of advantages and drawbacks could be complementary

-       FIG 4 : did you mean “heat treatment” instead of “heat curing”?

-       Lines 996-1002 : reference missing?

Comments on the Quality of English Language

Please see suggestions to authors

Author Response

Comments and Suggestions for Authors

 The authors review orodispersible films with specific focus on formulation, methods of preparation, evaluation, clinical studies and patent-based innovations and a table of commercialized products as examples.

Some parts of the subject can be found in other reviews. The strength of this review is to bring together various aspects, and to summarize the literature available in tables and figures, which are helpful for the reader.

 Answer: We express our gratitude to the reviewer for providing invaluable feedback that undoubtedly enhances the overall quality of the manuscript.

To further improve the manuscript, here are my suggestions :

-       Lines 40-50 : to be consistent with what is said after for orodispersible films (line 65), please add local effect (eg : “enabling rapid absorption and delivery of the drug into the bloodstream” or rapid local effect)

Answer: Based on the invaluable input from the reviewer, the sentence in the manuscript has been revised (lines 61-62).

-       In chapter Formulation components: a general table/figure, for example target product profile or generic composition with expected properties, could be helpful and complementary to the tables 1 and 2

Answer: We highly appreciate the insightful feedback from the reviewer and a comprehensive table (Table 2) that integrates the target product profile has been incorporated.

-       Table 3 (general): the table make sense but it’s organization could be improved : for instance by making sub-sections (eg mechanical properties, surface characterization, release properties…).

Answer: We express our gratitude to the reviewer for providing constructive feedback that will undoubtedly enhance the overall quality of the manuscript. In response to the comments, we have modified Table 3 (Now Table 4) by introducing distinct subsections to enhance the organizational structure of the table.

-       Table 3 - Specific comments : for some items, the content of the column “Evaluation parameters” or “Range units” seems confusing – item "Thickness, area and weight variation" has for instance as evaluation parameter Content uniformity (which should be a separate item, but does not characterize the thickness, area and weight variation?). For the item "Taste perception", it is only about friction : what about the taste and its acceptability (eg bitterness scores etc)?

Answer: We value the comprehensive review of the manuscript by the expert reviewers. Taking into account the valuable feedback, we have thoroughly revised Table 3 (Now Table 4), addressing all the concerns raised by the expert reviewers.

-       Figures for illustration of techniques are helpful – a comparison table of advantages and drawbacks could be complementary

Answer: We express our gratitude to the reviewer for providing valuable insights. In response to the feedback, we have included the images to illustrate various in vitro techniques (Figures 7-9) and constructed a comparison table outlining the advantages and drawbacks of orodispersible film drug delivery systems (New Table 1).

-       FIG 4 : did you mean “heat treatment” instead of “heat curing”?

Answer: We apologize for the oversight; the term "heat curing" in the figure 4 has been substituted with "heat treatment."

-       Lines 996-1002 : reference missing?

Answer: Apologies for the error; the missing reference has now been included in the revised manuscript (Lines 986-992).

Reviewer 2 Report

Comments and Suggestions for Authors

The review article entitled ‘Orodispersible Films: Current Innovations and Emerging Trends’ is a welcome addition to the literature, encompassing several poignant aspects in the research field, with a comprehensive overview of the recent studies and clinical advances.

The abstract could be improved, as it reads more like a formulaic description of what is contained in the review. Whilst this isn’t “wrong”, per se, I just feel that the purpose of the review could be better described for the reader. The title talks about innovations and emerging trends, so that should also be the focus of the abstract, whilst highlighting their importance in overcoming challenges and limitations of ODFs (this is done, in parts, but too sporadically).

The opening 2 section headings could certainly be more concise, as much of the information is repeated in some way later on (or, indeed, within those sections).

This trend continues throughout, as the information can often be more detailed than is required and lose some focus. The section on plasticisers certainly needs reviewing to bring the focus back to ODFs.

Overall, this is a valuable review article, which would be acceptable for publication following the amendments primarily focussed on the organisation and coherency of the information contained within. Several comments have been annotated on the attached file.

Comments on the Quality of English Language

Generally fine, only minor amendments that ought to be picked up on proof-reading.

Author Response

Comments and Suggestions for Authors

The review article entitled ‘Orodispersible Films: Current Innovations and Emerging Trends’ is a welcome addition to the literature, encompassing several poignant aspects in the research field, with a comprehensive overview of the recent studies and clinical advances.

The abstract could be improved, as it reads more like a formulaic description of what is contained in the review. Whilst this isn’t “wrong”, per se, I just feel that the purpose of the review could be better described for the reader. The title talks about innovations and emerging trends, so that should also be the focus of the abstract, whilst highlighting their importance in overcoming challenges and limitations of ODFs (this is done, in parts, but too sporadically).

Answer: We would like to convey our sincere thanks for the time and suggestions by the potential reviewer, which would surely improve the quality of the article. Subsequently, we have meticulously revised the abstract section of the manuscript, incorporating innovations, recent trends in ODF, and enhancing overall readability for the reader.

The opening 2 section headings could certainly be more concise, as much of the information is repeated in some way later on (or, indeed, within those sections).This trend continues throughout, as the information can often be more detailed than is required and lose some focus. The section on plasticisers certainly needs reviewing to bring the focus back to ODFs.

Answer: We fully agree with the esteemed reviewer regarding the elaboration of certain sections in the manuscript. Taking into account the valuable inputs, those sections including plasticizers were significantly condensed to enhance conciseness and improve readability for the readers.

Overall, this is a valuable review article, which would be acceptable for publication following the amendments primarily focussed on the organisation and coherency of the information contained within. Several comments have been annotated on the attached file.

Answer: We are happy to see the positive comments from the eminent reviewer. We would also like to thank the reviewer for raising the concerns, which would improvise the quality of our manuscript. The answers for each question in the pdf file are replied in the same document and the modifications are incorporated in the revised manuscript (word doc).

Comments on the Quality of English Language

Generally fine, only minor amendments that ought to be picked up on proof-reading.

Reviewer 3 Report

Comments and Suggestions for Authors

This review is well-written, well-organized, and informative. The review gives an excellent overview of all relevant aspects related to ODFs. It is therefore a good contribution to Pharmaceutics. I have a few minor comments and suggestions for possible improvements.

Abstract

Line 15 ...its potential -> ...their potential

Line 16-17: grammar deserves attention (detailed description....are (!) discussed.

Line 23: Add in what respect ADFs were evaluated in vitro.

Introduction

I would add at the end a clear aim / scope of the study.

What should be added as well, in my opinion, is the approach followed to write this review. How was the literature collected, from which databases, etc.

3.1

Heading: replace Drugs by API (more consistent with the text).

3.2

Heading: Polymers instead of Polymer.

6. In Vitro and In Vivo Evaluation Methods 

I would add ...to assess quality.

Please provide requirements from major pharmacopoieas in this section (USP, Ph. Eur. for instance). Are ODFs explicitly mentioned in a monograph? Or should requirements for other oral dispersible dosage forms be applied?

7. Clinical Translation and Future Prospects

In this section I miss a discussion about limitations of ODFs, which are definitely there. Think of limited API load, taste, uniformity of content. There is always another side of the medal.

Comments on the Quality of English Language

The English is good, as far as I can judge as a non-native speaker. Some mistakes to be corrected, but this can be easily done during the production process.

Author Response

Comments and Suggestions for Authors

This review is well-written, well-organized, and informative. The review gives an excellent overview of all relevant aspects related to ODFs. It is therefore a good contribution to Pharmaceutics. I have a few minor comments and suggestions for possible improvements.

Answer: Thank you for your constructive feedback on our review. We appreciate your positive comments regarding the overall quality and organization of the manuscript, as well as your acknowledgment of its informative content. We have carefully considered your comments and suggestions for possible improvements. We will diligently address these points to further enhance the clarity and completeness of our review.

Abstract

Line 15 ...its potential -> ...their potential

Answer: Said mistake was corrected in the manuscript (line 15).

Line 16-17: grammar deserves attention (detailed description....are (!) discussed.

Answer: Thank you for pointing out these grammatical mistakes. The sentence has now been revised in the manuscript (lines 16-17).

Line 23: Add in what respect ADFs were evaluated in vitro.

Answer: Thank you for pointing out this error. For clarity, the sentence was modified in the revised manuscript (lines 22-23).

Introduction

I would add at the end a clear aim / scope of the study.

What should be added as well, in my opinion, is the approach followed to write this review. How was the literature collected, from which databases, etc.

Answer: We express our gratitude to the reviewer for offering valuable insights that contribute to the manuscript's overall quality. The introduction section now incorporates a well-defined objective for this narrative review. Furthermore, we have incorporated details about the scientific approach employed in preparing this review (lines 75-87).

3.1

Heading: replace Drugs by API (more consistent with the text).

Answer: Thank you for bringing this to my attention. We have replaced the term "drugs" with "API" in the heading.

3.2

Heading: Polymers instead of Polymer.

Answer: Thank you for highlighting this grammatical error. We have made the correction in the heading by replacing the term "Polymer" with "Polymers."

  1. In Vitro and In Vivo Evaluation Methods

I would add ...to assess quality. Please provide requirements from major pharmacopoieas in this section (USP, Ph. Eur. for instance). Are ODFs explicitly mentioned in a monograph? Or should requirements for other oral dispersible dosage forms be applied?

Answer: Thank you for the invaluable feedback from the reviewer. In vivo evaluation methods, along with adherence to pharmacopoeial requirements where applicable, have been included in Table 4. ODFs were initially incorporated in the seventh edition of the European Pharmacopoeia in 2012. However, as of now, ODFs have not been formally established as an official monograph in widely recognized pharmacopoeias, such as the USP.

  1. Clinical Translation and Future Prospects

In this section I miss a discussion about limitations of ODFs, which are definitely there. Think of limited API load, taste, uniformity of content. There is always another side of the medal.

Answer: Acknowledgments to the reviewer for the keen observation regarding the omission of ODF drawbacks under the section on clinical translation and future prospects. The revised article now incorporates a concise summary addressing the limited API load, taste, and uniformity of content (lines 1047-1056). Additionally, a table comparing the advantages and drawbacks of orodispersible film drug delivery systems has been added to the manuscript (Table 1).

Comments on the Quality of English Language

The English is good, as far as I can judge as a non-native speaker. Some mistakes to be corrected, but this can be easily done during the production process.

Answer: The whole article is thoroughly reviewed by one of our English-speaking colleagues and edited for language style, grammar, spelling, and appropriate modifications are incorporated in the revised manuscript.
